# Dimensional evolution of charge mobility and porosity in covalent organic frameworks

Shuai Fu [1,5], Xiao Li[2,5], Guanzhao Wen[1], Yunyu Guo[2], Matthew A. Addicoat [3], Mischa Bonn [1], Enquan Jin [1,2] ✉, Klaus Müllen [1] ✉ & Hai I. Wang [1,4] ✉

Covalent organic frameworks are an emerging class of covalently linked polymers with programmable lattices and well-defined nanopores. Developing covalent organic frameworks with both high porosity and excellent charge transport properties is crucial for widespread applications, including sensing, catalysis, and organic electronics. However, achieving the combination of both features remains challenging due to the lack of overarching structure-property correlations. Here, we report a strategy toward covalent organic frameworks with tunable dimensionality. The concept relies on splicing one-dimensional charge-conducting channels to form extended networks with tailorable substitution patterns. Such dimensional evolution and substitution control enable fine-tuning of electronic band structure, charge mobility, and porosity. According to surface-area characterization, high-frequency terahertz photoconductivity measurements, and theoretical calculations, the transition from one-dimensional to para-linked two-dimensional networks furnishes a substantial increase in surface area and a decrease in local charge mobility. The latter feature is assigned to substitution-induced electronic band flattening. A subtle balance of surface area (947 $m^2 \cdot g^{-1}$) and local charge mobility (49 ± 10 $cm^2 \cdot V^{-1} \cdot s^{-1}$) is achieved through the rational design of meta-linked analogs with mixed one-dimensional and two-dimensional superior nature. This work provides fundamental insights and new structural knobs for the design of conductive covalent organic frameworks.

Covalent organic frameworks (COFs) incorporate organic units into long-range ordered lattices through reversible and dynamic covalent reactions. The chemical diversity of ligands provides degrees of freedom for designing linkages[1–5], topological networks[6–10], morphologies[11–13], planarity[14], stacking modes[15–17], and dimensionalities[18–20]. Exploiting their programmable nanopores and periodic skeletons has inspired two seemingly independent research targets: (i) porous COFs with high surface area, abundant active adsorption sites, and customizable nanopores for efficient mass separation[21,22], energy storage[23,24], and catalysis[25,26]; and (ii) conductive

COFs with dispersive energy bands[27,28], tunable bandgaps[29,30], and extended electronic states[31,32] for organic electronics[33], photovoltaics[34], and photodetection[35,36]. Along the former direction, dual-pore COFs with hierarchical porosity were synthesized by tessellating different topological patterns into the same skeleton to form a periodically ordered distribution of micropores and mesopores[37]. Furthermore, large-pore COFs with mesopore pore sizes up to 10 nm have recently been reported by varying the conformational rigidity, planarity, and local polarity of the monomers[38]. On the other hand, attempts have been made to improve the charge transport properties

[1]Max Planck Institute for Polymer Research, Ackermannweg 10, Mainz, Germany. [2]State Key Laboratory of Inorganic Synthesis and Preparative Chemistry, College of Chemistry and International Center of Future Science, Jilin University, Changchun, P.R. China. [3]School of Science and Technology, Nottingham Trent University, Clifton Lane, Nottingham, UK. [4]Nanophotonics, Debye Institute for Nanomaterials Science, Utrecht University, Princetonplein 1, Utrecht, The Netherlands. [5]These authors contributed equally: Shuai Fu, Xiao Li. ✉e-mail: enquanjin@jlu.edu.cn; muellen@mpip-mainz.mpg.de; h.wang5@uu.nl

of COFs through chemical doping[39,40], developing π-conjugated linkages[41,42], and using extended π-conjugated building blocks[43,44]. On this basis, recent spectroscopic studies have observed Drude-type delocalized charge transport with exceptionally high charge mobility comparable to conventional inorganic semiconductors in highly conjugated, crystalline COF films[27,45].

Despite these advances, COFs with both decent porosity and charge mobility remain scarce, yet they would be highly desirable for applications such as chemical sensing, catalysis, and batteries. One potential strategy to transcend this hurdle is dimensional engineering. This could be achieved by first connecting zero-dimensional (0D) small molecules to one-dimensional (1D) chains, which were further assembled and extended into two-dimensional (2D) or three-dimensional (3D) networks. Seminal work by Yaghi et al. [19] linked molecules and 1D ribbons into COFs in a stepwise manner. The success of this strategy inspired us to go one step further by manipulating the electronic coupling between the constituting 1D channels. Furthermore, while the pore size and distribution can be comprehensively characterized by nitrogen-sorption isotherm and Brunauer-Emmett-Teller (BET) theory, investigating the intrinsic charge transport properties of COFs through conventional device measurements remains challenging due to their limited grain size and problems of electrode penetration and large contact resistance.

Herein, we construct perylene-based 1D COF (1D Pery-COF) through the polycondensation of 4,4′,4″,4‴-(perylene-2,5,8,11-tetrayl) tetraaniline (PTTA) and [1,1′:3′,1″-terphenyl]−4,4″-dicarbaldehyde (TPDA), producing eclipsed AA stacking lattices with a surface area of 370 $m^2 \cdot g^{-1}$ and $\mu_{loc}$ of $66 \pm 14$ $cm^2 \cdot V^{-1} \cdot s^{-1}$. Stitching 1D Pery-COF via their para-positions can be achieved by replacing TPDA with 4′,5′-bis(4-formylphenyl)-[1,1′:2′,1″-terphenyl]−4,4″-dicarbaldehyde (FTDA) under optimized solvothermal conditions. The resulting para-linked perylene-based 2D COF (2D PL-Pery-COF) exhibits an enhanced surface area of 944 $m^2 \cdot g^{-1}$ thanks to spatial expansion, while $\mu_{loc}$ is reduced to $21 \pm 4$ $cm^2 \cdot V^{-1} \cdot s^{-1}$. Density functional tight binding (DFTB) calculations of the band structure rationalize the observed reduced carrier mobility by an emerging band flattening effect. This is induced by para-substitution and the enhanced π−π interaction of the stacked layers of the 2D extended structure. Inspired by this, we further designed meta-linked perylene-based 2D COF (2D ML-Pery-COF) with quasi-independently oriented 1D charge-conducting channels toward the desired balance between surface area and mobility. Based on the polycondensation of PTTA and 5′,5‴-bis(4-formylphenyl)-[1,1′:3′,1″:3″,1‴-quaterphenyl]−4,4‴-dicarbaldehyde (FQDA), the resulting meta-linked perylene-based 2D COF exhibits a large surface area of 947 $m^2 \cdot g^{-1}$ and a decent $\mu_{loc}$ of $49 \pm 10$ $cm^2 \cdot V^{-1} \cdot s^{-1}$. This work not only demonstrates COFs with different dimensionalities and controllable stitching strategies but also elucidates the role of dimensional evolution and substitution patterns in porosity and charge transport.

## Results
### Synthesis and characterization of perylene-based COFs with different dimensionalities
Highly crystalline 1D Pery-COF (Fig. 1a) and 2D PL-Pery-COF (Fig. 1d) were synthesized by the polycondensation of PTTA with TPDA and FTDA under optimized solvothermal conditions, respectively (Methods). Using acetic acid solution (HAc, 6 M) as the catalyst and heating at 120 °C for 72 h, the target 1D Pery-COF and 2D PL-Pery-COF were formed with yields of 77% and 86%, respectively. According to PXRD, 1D Pery-COF exhibited distinguishable peaks at 3.92°, 4.64°, 6.34°, 7.84°, 11.33°, and 25.65°, corresponding to the (110), (200), (020), (220), (500), and (001) facets (Fig. 1b); 2D PL-Pery-COF displayed diffraction peaks at 5.62°, 9.38°, 11.42°, 17.12°, and 21.53°, corresponding to the (110), (300), (220), (330), and (001) planes, respectively (Fig. 1e). After geometry optimization through DFTB calculations and Pawley refinement (Methods) of the simulated crystal models, the final cell

parameters were determined to be $a = 39.3410$ Å, $b = 27.8420$ Å, $c = 3.5950$ Å, $\alpha = \beta = \gamma = 90°$ with high symmetrical space group of $P1$ ($R_p = 1.09\%$ and $R_{wp} = 1.67\%$) for 1D Pery-COF (Fig. 1c); $a = 29.3042$ Å, $b = 17.9815$ Å, $c = 4.1240$ Å, $\alpha = \beta = \gamma = 90°$ with $P222$ ($R_p = 2.91\%$ and $R_{wp} = 4.91\%$) for 2D PL-Pery-COF (Fig. 1f). The simulated AA-stacking patterns matched well with the experimental ones, confirming that both 1D and 2D PL-Pery-COFs adopted eclipsed AA stacking arrangements with interlayer distances of 3.6 Å (Supplementary Fig. 1a) and 4.0 Å (Supplementary Fig. 1b), while staggered AB stacking modes could not. Fourier transform infrared (FTIR) spectra of 1D Pery-COF and 2D PL-Pery-COF revealed the characteristic C=N stretching vibration at 1623 $cm^{-1}$ via polycondensation (Supplementary Fig. 2a, b). Solid-state cross-polarization/magic-angle spinning (CP-MAS) $^{13}C$ nuclear magnetic resonance (NMR) of 1D Pery-COF and 2D PL-Pery-COF displayed the C=N carbon peaks at 158 ppm. By contrast, the resonance signal at 190 ppm originating from the aldehyde carbon entirely disappeared in both COFs (Supplementary Fig. 3a, b), demonstrating the completeness of the condensation reaction. Elemental analysis results matched the theoretical formula (Supplementary Table 1). Scanning electron microscopy (SEM) images revealed that 1D Pery-COF and 2D PL-Pery-COF were rod-like microcrystals (Supplementary Fig. 4). According to thermogravimetric analysis (TGA), both of them showed high stability up to 450 °C under nitrogen atmosphere (Supplementary Fig. 5). The nitrogen-sorption isotherms of activated 1D Pery-COF and 2D PL-Pery-COF at 77 K revealed type IV patterns. The BET surface areas of 1D Pery-COF and 2D PL-Pery-COF were calculated to be 370 and 944 $m^2 \cdot g^{-1}$, respectively. Furthermore, their pore-size distributions (PSD) were evaluated to be 1.6 and 1.5 nm using the quenched solid density functional theory (QSDFT) model, in good agreement with the simulated crystalline structures (Fig. 2a and Supplementary Fig. 6a, b). In addition, high-resolution transmission electron microscopy (HR-TEM) demonstrated the existence of rhombus-shaped pores (Supplementary Fig. 7).

To fully exploit the substitution pattern of 1D Pery-COF, we developed a meta-linked perylene-based COF, termed 2D ML-Pery-COF (Fig. 1g). The 2D ML-Pery-COF largely preserves the 1D conducting nature thanks to the weak electronic coupling between the constituting 1D channels, although the network is spatially extended in a 2D meta-linked manner. According to the PXRD measurement (Fig. 1h), 2D ML-Pery-COF showed diffraction peaks at 3.89°, 5.01°, 7.87°, 10.04°, 15.01°, and 24.25°, corresponding to the (010), (110), (020), (220), (330), and (001) facets. After DFTB calculation and Pawley refinement, 2D ML-Pery-COF exhibited AA stacking mode (Fig. 1i) with optimized cell parameters of $a = 21.7971$ Å, $b = 29.6484$ Å, $c = 3.8798$ Å, $\alpha = \beta = \gamma = 90°$ with high symmetrical space group of $P1$ ($R_p = 1.29\%$ and $R_{wp} = 2.07\%$) and interlayer distance of 3.8 Å (Supplementary Fig. 1c).

### Dimensional evolution of porosity, optical, and electronic properties
The nitrogen sorption measurement recorded a BET surface area of 947 $m^2 \cdot g^{-1}$ (Fig. 2a, blue curve), and the PSD calculation revealed a pore size of 1.5 nm (Supplementary Fig. 6c). The resulting porosity of 2D ML-Pery-COF is comparable to traditional 2D imine-linked COFs[46,47]. Moreover, FTIR spectra (Supplementary Fig. 2c), TGA, CP-MAS spectrometer (Supplementary Fig. 3c), and SEM images (Supplementary Fig. 4c) of 2D ML-Pery-COF all exhibited similar characteristics to those of 2D PL-Pery-COF. The KM reflectance spectra revealed that the absorption onset wavelengths of 1D Pery-COF, 2D ML-Pery-COF, and 2D PL-Pery-COF were 571, 537, and 523 nm (Fig. 2b), respectively. Tauc plot analysis of 1D Pery-COF, 2D ML-Pery-COF, and 2D PL-Pery-COF revealed their semiconducting nature with indirect optical bandgaps of 2.17, 2.31, and 2.37 eV (Fig. 2c), respectively. To understand the increase in bandgap with dimensionality, absorption features of 1D Pery-COF, 2D ML-Pery-COF, and 2D PL-Pery-COF were simulated using Materials Studio (version 4.4, see details in Supplementary

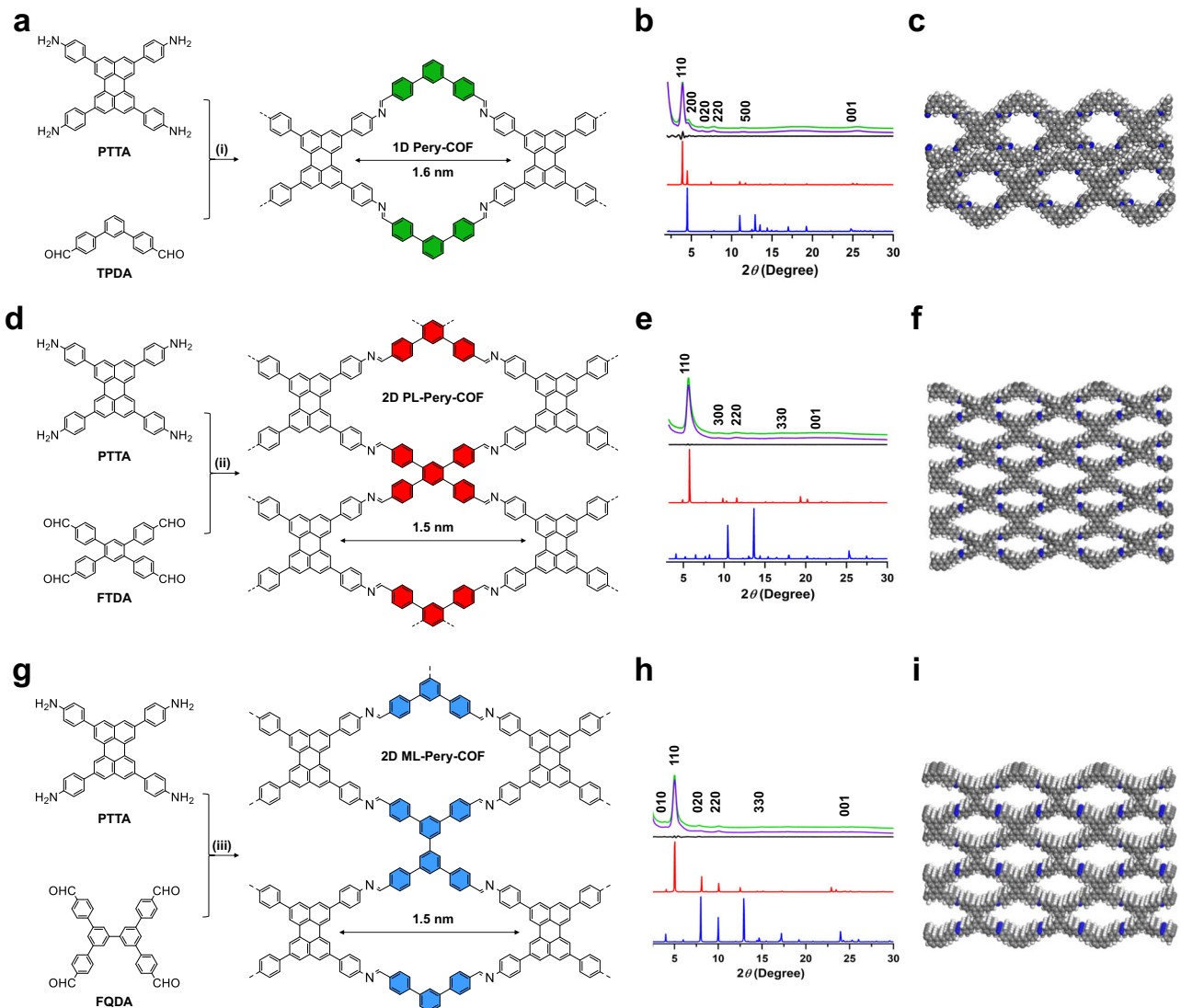

**Fig. 1 | Synthesis of perylene-based COFs. a, d, g** Synthetic routes of (**a**) 1D Pery-COF, (**d**) 2D PL-Pery-COF, and (**g**) 2D ML-Pery-COF. **b, e, h** Powder X-ray diffraction (PXRD) of experimentally observed (green curves), Pawley refined (purple curves), their difference (black curves), and simulated AA (red curves) and AB stacking (blue curves) patterns of (**b**) 1D Pery-COF, (**e**) 2D PL-Pery-COF, and (**h**) 2D ML-Pery-COF. **c, f, i** DFTB-optimized crystalline structures viewed along the pseudo-quadratic pore of multilayered (**c**) 1D Pery-COF, (**f**) 2D PL-Pery-COF, and (**i**) 2D ML-Pery-COF (gray, carbon; blue, nitrogen; white, hydrogen). Solvothermal conditions: (i) HAc (6 M), *o*-dichlorobenzene (ODCB)/*n*-butanol (*n*-BuOH) (*v/v* = 4/1), 120 °C, 72 h, 77%; (ii) HAc (6 M), ODCB/*n*-BuOH (*v/v* = 3/1), 120 °C, 72 h, 86%; (iii) HAc (6 M), ODCB/*n*-BuOH (*v/v* = 3/1), 120 °C, 72 h, 82%.

Information)[48]. The simulated results closely matched the experimental ones (Supplementary Fig. 8). The bandgaps of COFs are primarily determined by the extent of π-conjugation within their polymeric skeletons. In the 2D PL-Pery-COF, the twist angle between two adjacent benzene rings in the FTDA unit is 40.66°, larger than the 24.12° twist angle between the meta-substituted benzene and the central benzene ring in the TPDA unit of 1D Pery-COF (Supplementary Fig. 9). Consequently, the more non-planar configuration of the FTDA unit in the 2D PL-Pery-COF hinders the extension of π-conjugation, leading to a broader bandgap and a blue-shifted absorption feature. The spatial extent of the highest occupied molecular orbital (HOMO) and the lowest unoccupied molecular orbital (LUMO) of 1D Pery-COF, 2D ML-Pery-COF, and 2D PL-Pery-COF were calculated using the periodic DFTB structures (Supplementary Table 2). The HOMO is located on the perylene moiety for all COFs, while the LUMO is mostly localized within the perylene moiety in 1D Pery-COF and 2D ML-Pery-COF (Supplementary Fig. 10). In contrast, the LUMO of 2D PL-Pery-COF is confined in the phenylene linker. Notably, the calculated electronic

band structure of 2D ML-Pery-COF displays similar band dispersion to 1D Pery-COF at the energy band extrema, whereas the band dispersion of 2D PL-Pery-COF at the energy band extrema is much smaller (Figs. 2d–f). These results indicate that (i) 1D Pery-COF forms directional charge-conducting channels with a relatively small electron-hole reduced effective mass ($m^*$, see computational details in Methods); and (ii) para-substitution has a more significant effect on electronic band flattening than meta-substitution. This is manifested by the flattening of the valence band, resulting in larger hole masses as the dimensionality increases. Overall, the simulation indicates an enhanced charge carrier localization and, thus, a decreased carrier mobility with increasing the dimensionality of Pery-COFs.

**Dimensional evolution of charge transport properties**

The effects of dimensional evolution and substitution patterns on charge transport properties were explored by contact-free time-resolved terahertz spectroscopy (TRTS). In TRTS measurements, an ultrashort 3.1 eV laser pulse optically injects charge carriers in the

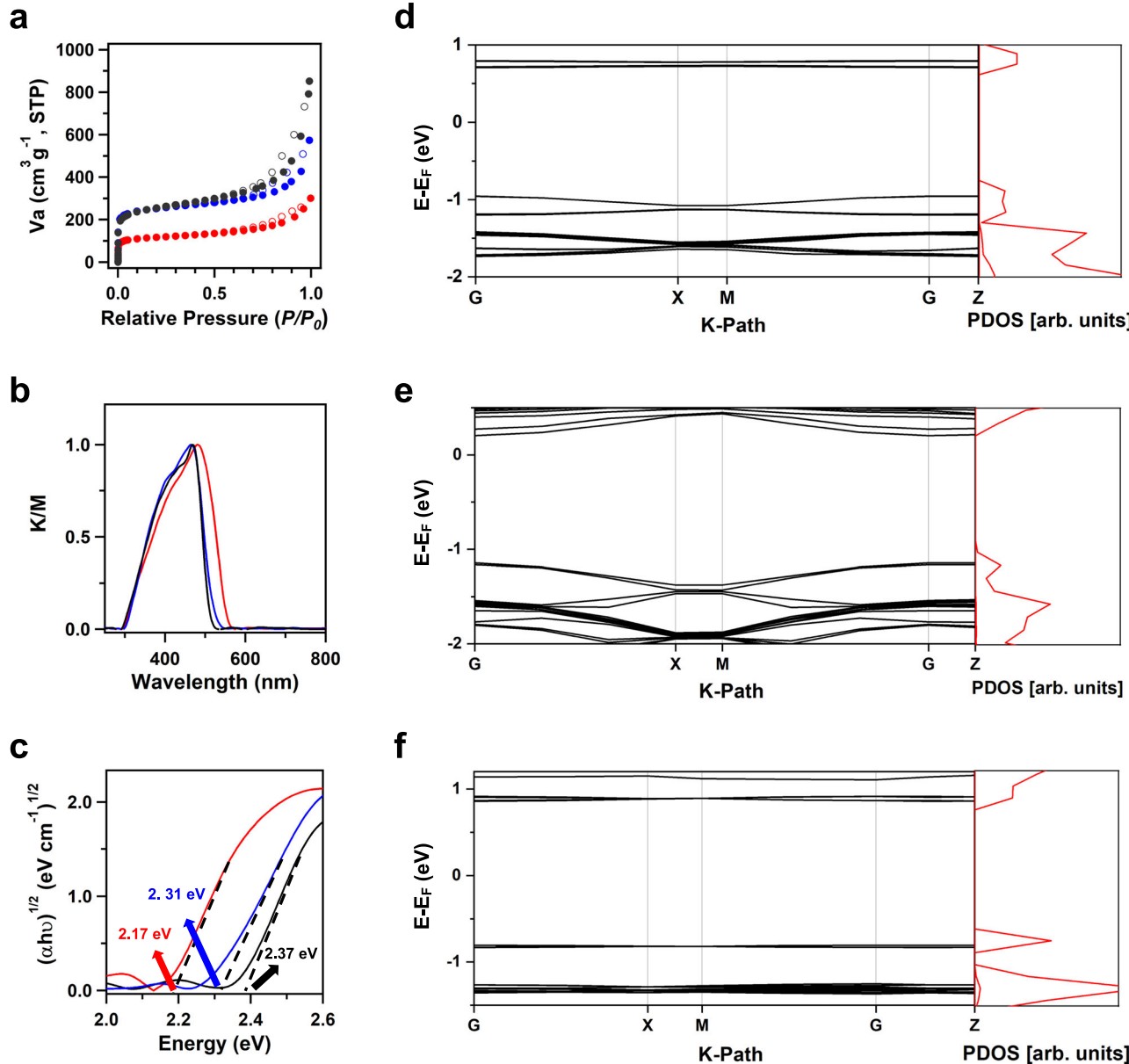

**Fig. 2 | Porosity, optical properties, and calculated electronic band structures of perylene-based COFs. a** Nitrogen sorption isotherm curves of 1D Pery-COF (red dots), 2D ML-Pery-COF (blue dots), and 2D PL-Pery-COF (black dots). **b** Solid-state UV/vis diffuse reflectance spectra of 1D Pery-COF (red curve), 2D ML-Pery-COF (blue curve), and 2D PL-Pery-COF (black curve). **c** Tauc plots showing the bandgaps determined from the Kubelka-Munk-transformed (KM) reflectance spectra of 1D Pery-COF (red curve), 2D ML-Pery-COF (blue curve), and 2D PL-Pery-COF (black curve). **d–f** Electronic band structures and projected density of states (PDOS) of (**d**) 1D Pery-COF, (**e**) 2D ML-Pery-COF, and (**f**) 2D PL-Pery-COF.

perylene-based COFs via above-bandgap excitation. The pump-induced conductivity changes are then probed with a freely propagating single-cycle THz electric field with a duration of ~1 ps (Fig. 3a). The transient nature of the THz pulse allows access to microscopic charge transport properties on length scales of tens of nm, providing a powerful method to study the intrinsic electrical properties of a variety of nanomaterials[45,49]. TRTS relies on the interaction between the pulsed terahertz (THz) electrical field and the photogenerated charge carriers[50]. The relative attenuation of the THz electric field ($-\Delta E/E$) by the absorption of charge carriers is proportional to the THz photoconductivity ($\Delta\sigma$); the $\Delta\sigma$ divided by absorbed photon density ($\Delta\sigma/N_{abs}$) further provides a direct comparison of charge mobility. As shown in Fig. 3b, the rapid sub-ps rise in photoconductivity dynamics originates from the photoinjection of conductive carriers, while the subsequent ps-level decay can be tentatively attributed to charge

localization or recombination[51]. The difference in $\Delta\sigma/N_{abs}$ indicates that their charge mobility follows 1D Pery-COF > 2D ML-Pery-COF > 2D PL-Pery-COF, in line with the simulation results.

The normalized frequency-resolved photoconductivity ($\Delta\sigma(\omega)$) near the maximum photoconductivity overlaps well (Fig. 3c), suggesting that 1D Pery-COF, 2D ML- Pery-COF, and 2D PL-Pery-COF have very similar charge transport behavior. The conductivity dispersion can be well captured by the Drude-Smith (DS) model. In the DS model, charge carriers experience preferential backscattering due to the presence of, e.g., grain boundaries or structural disorder, following Eq. (1):

$$\Delta\sigma(\omega) = \frac{\omega_p^2 \varepsilon_0 \tau}{1 - i\omega\tau}\left(1 + \frac{c}{1 - i\omega\tau}\right) \quad (1)$$

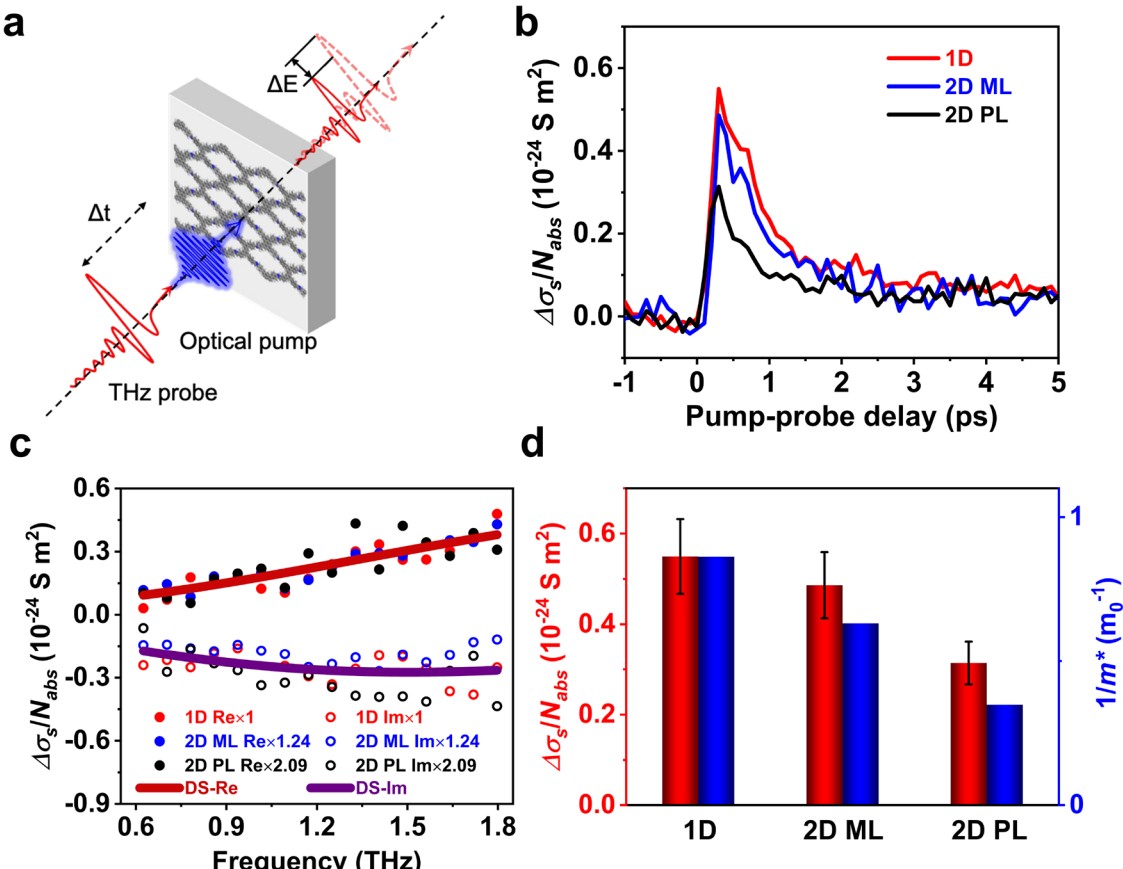

**Fig. 3 | Charge transport properties of perylene-based COFs. a** Schematic of time-resolved terahertz spectroscopy (TRTS). In the scheme, $\Delta t$ represents the relative time delay between the optical pump and the THz probe, and $\Delta E$ stands for the relative change of the THz electric field. **b** Time-resolved photoconductivity normalized by absorbed photon density ($\Delta\sigma/N_{abs}$) of perylene-based COFs. **c** Frequency-resolved photoconductivity ($\Delta\sigma(\omega)$) of perylene-based COFs measured at 0.5 ps after the maximum photoconductivity. The filled and empty circles represent the real and imaginary components of the complex THz photoconductivity of 1D Pery-COF (red), 2D ML-Pery-COF (blue, rescaled by 1.24 ×), and 2D PL-Pery-COF (black, rescaled by 2.09 ×). The red and purple solid lines correspond to the global fitting of the real and imaginary components of $\Delta\sigma(\omega)$ following the Drude-Smith (DS) model. **d** Correlation between the relative mobility ($\Delta\sigma/N_{abs}$) of perylene-based COFs and the inverse of their electron-hole reduced effective masses ($1/m^*$). The error bars represent the standard deviation, estimated from point-to-point variation.

where $\omega_p$ is the plasma frequency, $\varepsilon_0$ is the vacuum permittivity, $\tau$ is the DS charge scattering time, and $c$ (ranging from −1 to 0) is the backscattering parameter that limits the long-distance migration of charge carriers in the DC limit. By fitting $\Delta\sigma(\omega)$ to the DS model, we can infer $\tau$ of $43 \pm 4$ fs and $c$ parameter of − 0.97. In this sense, m* becomes the only factor that determines the local mobility $\mu_{loc}$ in the diffusive limit, following Eq. (2):

$$\mu_{loc} = e\tau/m^* \tag{2}$$

where e is the elementary charge. This argument is further supported by the direct correlation between $\mu_{loc}$ and $1/m^*$ shown in Fig. 3d, both decreasing as the dimensionality expands from 1D Pery-COF to 2D ML-Pery-COF to 2D PL-Pery-COF. Knowing $m^*$ from DFT calculations (see computational details in Methods and calculated carrier masses in Supplementary Table 2), $\mu_{loc}$ of 1D Pery-COF, 2D ML-Pery-COF, and 2D PL-Pery-COF are estimated to be $66 \pm 14$, $49 \pm 10$, and $21 \pm 4$ cm²·V⁻¹·s⁻¹, respectively. Utilizing the developed strategy, the achieved surface area and charge transport properties compare favorably with those of state-of-the-art conductive COFs (Supplementary Table 3). While the current experimental paradigm and analytical protocol limit the ability to isolate directional charge transport properties, theoretical calculations of directional carrier masses suggest that: (i) 1D Pery-COF exhibits low in-plane and out-of-

plane electron masses as well as a low hole mass, consistent with its highest charge mobility among the Pery-COF series; (ii) the most pronounced band-flattening effect observed in 2D PL-Pery-COF results from a combined effect of the substantial increase in hole mass and out-of-plane electron mass; and (iii) the high charge mobility maintained in 2D ML-Pery-COF relative to 1D Pery-COF is driven by a subtle balance between the increase in hole mass and in-plane electron mass and the decrease in out-of-plane electron mass.

## Discussion

Modulating the porosity and charge transport properties of COFs (Fig. 4) reveals an unexpected band-structure flattening effect that reduces the charge carrier mobility with increasing dimensionality. On the other hand, COFs with high surface area are favorable for gas/chemical adsorption, selection, and thus sensing. A delicate balance in charge transport and surface area by tuning the dimensionality and substitution pattern of COFs will be beneficial for catalysis or gas sensing, where abundant active adsorption sites and efficient charge transport are required.

To evaluate the broader applicability of our synthetic strategy and observed dimensionality-dependent properties, we synthesized a series of pyrene-based COFs with varying dimensionalities and substitution patterns, namely 1D Py-COF, 2D PL-Py-COF, and 2D ML-Py-COF, via Schiff-base condensation of 4,4',4'',4'''-(pyrene-1,3,6,8-tetrayl)

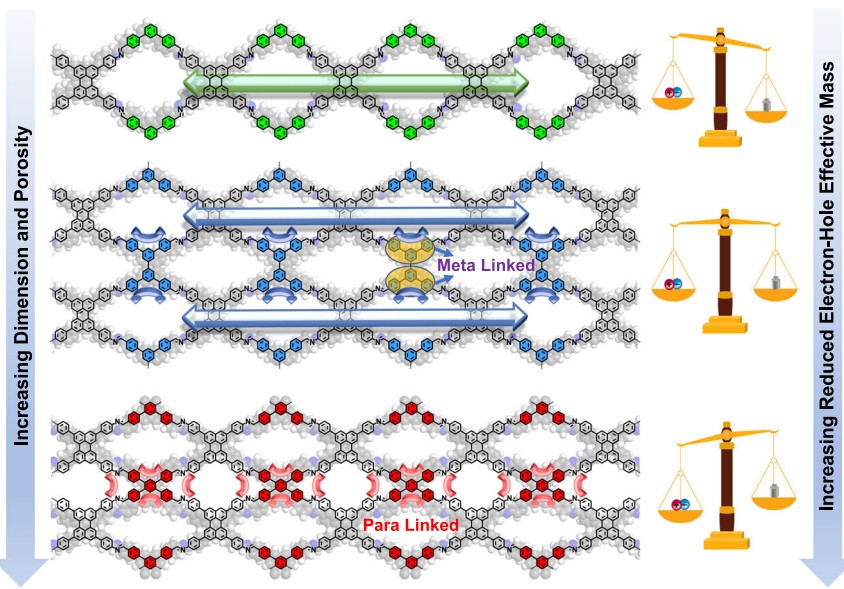

**Fig. 4 | Trade-off between porosity and reduced electron-hole effective mass driven by dimensional evolution.** Schematic picture showing that dimensional evolution increases charge carrier effective mass in 1D Pery-COF, 2D ML-Pery-COF, and 2D PL-Pery-COF.

tetraaniline (pyTTA) with TPDA, FTDA, FQDA, respectively (Fig. 5a, "Methods"). Based on systematic structural (Supplementary Figs. 11–14), optical, and THz photoconductivity characterizations, we observe that increasing the dimension from 1D to 2D results in (i) an increase in bandgap (from 2.15 to 2.28 eV; Supplementary Fig. 15) and (ii) a reduced photoconductivity amplitude per absorbed photon (Fig. 5b, c). Being in line with the results for Pery-COFs, the new series of pyrene-based COFs demonstrated again that the balance between surface area and charge mobility can be effectively optimized by judiciously tuning the dimensionality and substitution pattern, enhancing the applicability of the design strategy.

In summary, we have constructed two series of COFs with varying dimensionalities and substitution patterns based on perylene and pyrene units. Precise control of dimensionality and substitution patterns enables a systematic and fundamental understanding of how they modulate charge transport properties and porosity. We show that in the studied perylene-based COFs system, the trade-off between surface area (from 370 to 947 $m^2 \cdot g^{-1}$) and charge mobility (from $66 \pm 14$ to $21 \pm 4$ $cm^2 \cdot V^{-1} \cdot s^{-1}$) can be achieved by exploiting the spatial expansion and electronic band flattening effects, respectively. These findings enrich the toolkit for exploring the correlation between porosity and transport properties and unlock new degrees of freedom for designing conductive porous materials with on-demand features.

## Methods
### COF synthesis
As starting materials, 4′,5′-bis(4-formylphenyl)-[1,1′:2′,1″-terphenyl]−4,4″-dicarbaldehyde (FTDA) was purchased from Bidepharm at 98% purity and 4,4′,4″,4‴-(pyrene-1,3,6,8-tetrayl)tetraaniline (pyTTA) was purchased from Energy Chemical at 97% purity. The 1D Pery-COF, 2D ML-Pery-COF, 2D PL-Pery-COF, 1D Py-COF, 2D ML-Py-COF, and 2D PL-Py-COF were synthesized using the standard Schlenk line technique. Unless otherwise noted, all reagents were purchased from commercial chemical suppliers and used without further purification. Column chromatography separation was performed with silica gel (particle size 0.04−0.06 mm).

### Synthesis of PTTA
A reaction mixture containing 2,5,8,11-tetrakis(4,4,5,5-tetramethyl1,3,2-dioxaboranyl) perylene (1.0 g, 0.98 mmol), N-Boc-4-bromoaniline (1.12 g, 4.12 mmol), $K_2CO_3$ (1.63 g, 11.81 mmol), $Pd_2(dba)_3$ (270.06 mg, 0.3 mmol), and 2-Dicyclohexylphosphino-2′,6′-dimethoxybiphenyl (Sphos, 484.29 mg, 1.18 mmol) in 40 mL o-xylene and 10 mL $H_2O$ was stirred under a static argon atmosphere at 110 °C for 3 days. After cooling to room temperature, 20 mL MeOH was added to form a homogeneous suspension. The precipitate was collected by filtration and washed with methanol (MeOH) until the washings were clear. The greenish solid was purified by flash chromatography (dichloromethane (DCM)/EtOAc = 6/4), yielding the product as a greenish-yellow powder: tetra-tert-butyl (perylene-2,5,8,11-tetra-yltetrakis(benzene-4,1-diyl))tetracarbamate [Per(NHBoc)₄, 866.3 mg, 95% yield]. ¹H NMR (400 MHz, DMSO-$d_6$) δ 9.54 (s, 4H), 8.73 (s, 4H), 8.13 (s, 4H), 7.93 (s, 8H), 7.67 (s, 8H), 1.51 (s, 36H). A reaction mixture containing Per(NHBoc)₄ (1.00 g, 0.98 mmol) in 30 mL anhydrous DCM was treated with 10 mL trifluoroacetic acid. The resulting dark yellow solution was stirred at room temperature for 2 h. The reaction mixture was then neutralized by the slow addition of a saturated $Na_2CO_3$ solution in deionized water, leading to the precipitation of the product as a dark orange solid. The precipitate was collected by filtration, washed with 100 mL deionized water, and dried under a high vacuum, yielding PTTA as an orange powder (0.54 g, 88%). ¹H NMR (400 MHz, DMSO-$d_6$) δ 8.65 (s, 4H), 8.06 (s, 4H), 7.81 (d, J = 8.1 Hz, 8H), 6.93 (d, J = 8.0 Hz, 8H).

### Synthesis of TPDA
To a degassed solution of 1,3-dibromobenzene (0.50 g, 2.12 mmol), (4-formylphenyl) boronic acid (0.80 g, 5.34 mmol), and $K_2CO_3$ (1.13 g, 8.19 mmol) in 1,4-dioxane/$H_2O$ (100 mL/4 mL), Pd(PPh₃)₄ (100 mg) was added. The reaction mixture was heated at 90 °C overnight. After cooling to room temperature, the mixture was poured into $H_2O$ and filtered. The solid was washed with $H_2O$, ethanol, and acetone multiple times. The residue was then purified by column chromatography (n-hexane/DCM = 1: 1) to yield TPDA (0.80 g, 92% yield) as a white solid. TLC $R_f$ = 0.3 (n-hexane/DCM = 1: 2); ¹H NMR (400 MHz, Chloroform-d) δ 10.11 (s, 2H), 8.11 (s, 2H), 7.54 (d, J = 7.9 Hz, 5H), 7.38 (d, J = 7.7 Hz, 5H).

### Synthesis of FQDA
To a degassed solution of 3,3′,5,5′-tetrabromo-1,1′-biphenyl (1.00 g, 2.12 mmol), (4-formylphenyl) boronic acid (1.34 g, 8.94 mmol), and $K_2CO_3$ (3.52 g, 25.50 mmol) in 1,4-dioxane/$H_2O$ (100 mL/4 mL),

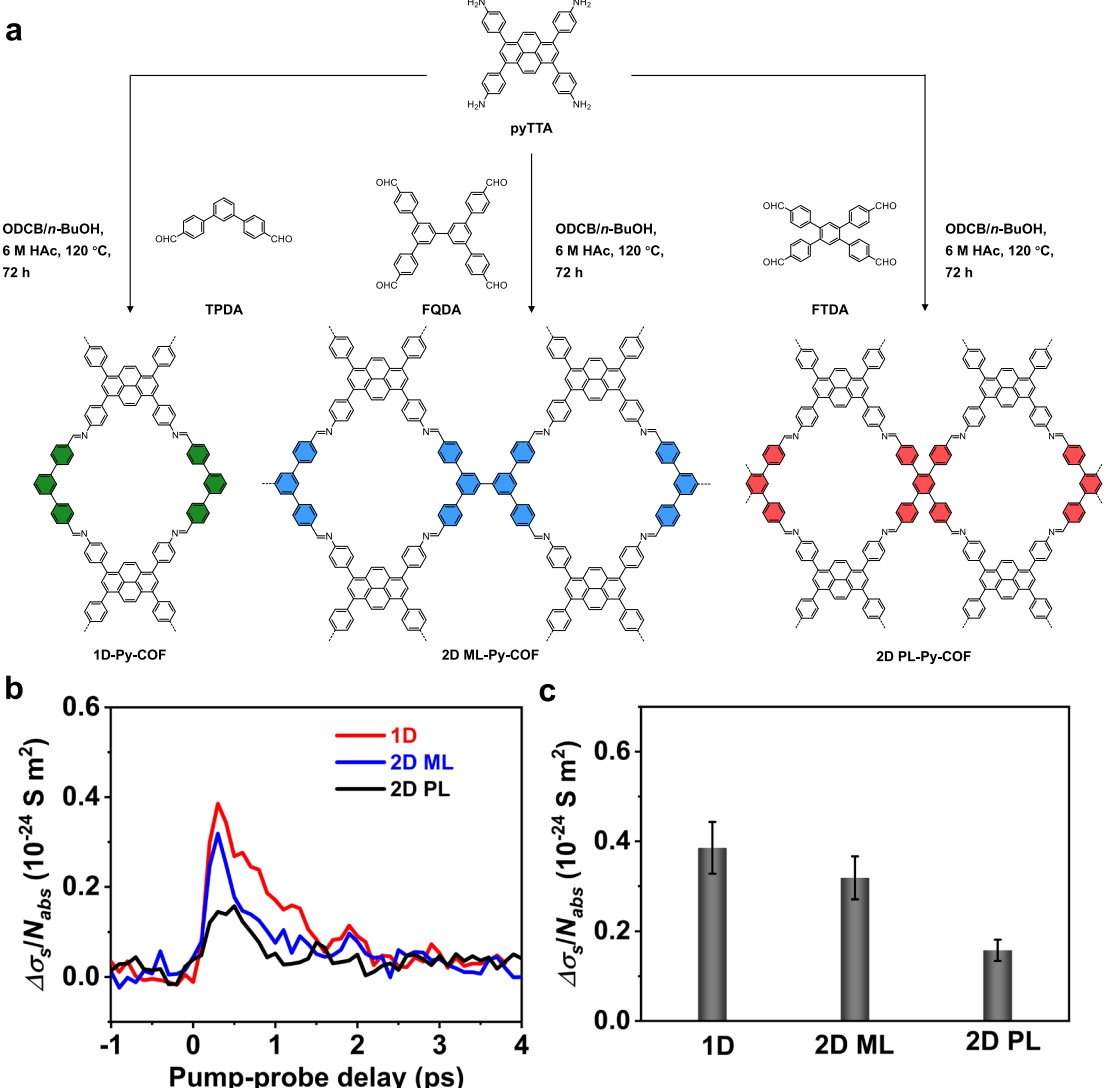

**Fig. 5 | Synthesis and charge transport properties of pyrene-based COFs.**
**a** Synthetic routes of 1D Py-COF, 2D-ML-Py-COF, and 2D-PL-Py-COF. **b** Time-resolved photoconductivity normalized by absorbed photon density ($\Delta\sigma/N_{abs}$) of pyrene-based COFs. **c** Comparison of the peak photoconductivity normalized by absorbed photon density ($\Delta\sigma/N_{abs}$) of pyrene-based COFs. The error bars represent the standard deviation, estimated from point-to-point variation.

Pd(PPh$_3$)$_4$ (200 mg, 0.17 mmol) was added. The reaction mixture was heated at 120 °C overnight. After cooling to room temperature, the mixture was poured into H$_2$O (150 mL) and filtered. The solid was washed with 1 M HCl, H$_2$O, ethanol, and acetone multiple times. The residue was purified by column chromatography (*n*-hexane/tetra-hydrofuran (THF) = 1: 2) to afford FQDA (0.80 g, 92% yield) as a white solid. TLC $R_f$ = 0.3 (*n*-hexane/THF = 1: 2); $^1$H NMR (400 MHz, CDCl$_3$, 298 K, ppm) $\delta$ 10.11 (s, 4H), 8.02 (d, $J$ = 6.56 Hz, 8H), 7.96 (s, 4H), 7.91 (d, $J$ = 6.92 Hz, 10H). HR MS (MALDI-TOF): m/z Calcd for C$_{40}$H$_{26}$O$_4$: 570.6440, found: 570.6443.

## Synthesis of 1D Pery-COF

A microwave tube (15 mL) was placed with PTTA (12.0 mg, 0.019 mmol) and TPDA (11.1 mg, 0.039 mmol), and a mixture of 1,2-dichlorobenzene (ODCB) (0.80 mL)/*n*-butanol (*n*-BuOH) (0.20 mL)/6 M HAc (0.10 mL) was added and sonicated for 5 min and degassed through three freeze-pump-thaw cycles before sealing under vacuum. The tube was sealed and heated at 120 °C for 72 h. After cooling to room temperature, the precipitate was filtered and washed with THF and acetone three times. Further purification was carried out by Soxhlet extraction in THF for 24 h and chloroform for 24 h. The powder was activated by

supercritical carbon dioxide and dried under vacuum at 60 °C for 12 h to produce 1D Pery-COF in a 77% isolated yield.

## Synthesis of 2D ML-Pery-COF

A microwave tube (15 mL) was placed with PTTA (12.0 mg, 0.019 mmol), FQDA (11.1 mg, 0.019 mmol), and a mixture of ODCB (0.75 mL)/*n*-BuOH (0.25 mL)/6 M HAc (0.10 mL) was added and soni-cated for 5 min and degassed through three freeze-pump-thaw cycles before sealing under vacuum. The tube was sealed and heated at 120 °C for 72 h. After cooling to room temperature, the precipitate was filtered and washed with THF and acetone three times. Further puri-fication was carried out by Soxhlet extraction in 1,4-dioxane for 24 h and acetone for 24 h. Then, the residue was activated by supercritical carbon dioxide and dried under vacuum at 60 °C for 12 h to produce 2D ML-Pery-COF in an 82% isolated yield.

## Synthesis of 2D PL-Pery-COF

A microwave tube (15 mL) was placed with PTTA (12.0 mg, 0.019 mmol), FTDA (9.6 mg, 0.019 mmol), and a mixture of ODCB (0.75 mL)/*n*-BuOH (0.25 mL)/6 M HAc (0.10 mL) was added and soni-cated for 5 min and degassed through three freeze-pump-thaw cycles

before sealing under vacuum. The tube was sealed and heated at 120 °C for 72 h. After cooling to room temperature, the precipitate was filtered and washed with THF and acetone three times. Further purification was carried out by Soxhlet extraction in 1,4-dioxane for 24 h and acetone for 24 h. Then, the residue was activated by supercritical carbon dioxide and dried under vacuum at 60 °C for 12 h to produce 2D PL-Pery-COF in an 86% isolated yield.

### Synthesis of 1D Py-COF

A pyrex tube (10 mL) was placed with 4,4′,4″,4‴-(pyrene-1,3,6,8-tetrayl) tetraaniline (pyTTA) (10.0 mg, 0.018 mmol) and TPDA (10.1 mg, 0.035 mmol). A mixture of ODCB (0.20 mL)/n-BuOH (0.80 mL)/6 M HAc (0.10 mL) was added, followed by sonication for 5 min and degassing through three freeze-pump-thaw cycles. The tube was sealed under vacuum and heated at 120 °C for 72 h. After cooling to room temperature, the precipitate was centrifuged and washed three times with THF and acetone. Further purification was carried out by Soxhlet extraction in THF for 24 h and acetone for 24 h. The resulting powder was activated using supercritical carbon dioxide and dried under vacuum at 60 °C for 12 h, affording 1D Py-COF in a 53% isolated yield.

### Synthesis of 2D ML-Py-COF

A pyrex tube (10 mL) was placed with pyTTA (10.0 mg, 0.018 mmol) and FQDA (10.0 mg, 0.018 mmol). A mixture of ODCB (0.80 mL)/n-BuOH (0.20 mL)/6 M HAc (0.10 mL) was added, followed by sonication for 5 min and degassing through three freeze-pump-thaw cycles. The tube was sealed under vacuum and heated at 120 °C for 72 h. After cooling to room temperature, the precipitate was centrifuged and washed three times with THF and acetone. Further purification was carried out by Soxhlet extraction in THF for 24 h and acetone for 24 h. The resulting residue was activated using supercritical carbon dioxide and dried under vacuum at 60 °C for 12 h, affording 2D ML-Py-COF in a 69% isolated yield.

### Synthesis of 2D PL-Py-COF

A pyrex tube (10 mL) was placed with pyTTA (12.0 mg, 0.021 mmol) and FTDA (10.4 mg, 0.021 mmol). A mixture of ODCB (0.20 mL)/n-BuOH (0.80 mL)/6 M HAc (0.10 mL) was added, followed by sonication for 5 min and degassing through three freeze-pump-thaw cycles. The tube was sealed under vacuum and heated at 120 °C for 72 h. After cooling to room temperature, the precipitate was centrifuged and washed three times with THF and acetone. Further purification was carried out by Soxhlet extraction in THF for 24 h and acetone for 24 h. The resulting residue was activated using supercritical carbon dioxide and dried under vacuum at 60 °C for 12 h, affording 2D PL-Py-COF in a 66% isolated yield.

### Basic characterization

Solution Nuclear Magnetic Resonance (NMR) spectra were recorded on a Bruker Avance NEO 400 MHz NMR spectrometer. Chemical shifts ($\delta$) were expressed in ppm relative to the deuterium solvents. Coupling constants ($J$) were recorded in Hertz. The solid-state $^{13}C$ cross-polarization/magic-angle spinning (CP-MAS) NMR spectra were collected with a Bruker AVANCE III 400 WB spectrometer. High-resolution mass spectra (HR-MS) were recorded on a Bruker Reflex II-TOF spectrometer by matrix-assisted laser decomposition/ionization (MALDI) using 7,7,8,8-tetracyanoquinodimethane (TCNQ) as matrix calibrated with poly(ethylene glycol). The Kubelka-Munk-transformed (KM) reflectance spectra were measured on a Perkin-Elmer Lambda 900 spectrometer. Fourier transform infrared (FTIR) spectroscopy was recorded on a Bruker TENSOR II FTIR spectrometer with KBr pellets. Each sample was conducted with a scan number of 64 and the background was subtracted. Scanning electron microscope (SEM) images were recorded on a LEO Gemini 1530 (Carl Zeiss AG, Germany).

High-resolution transmission electron microscopy (HR-TEM) images were taken with a JEOL2100F. Thermal gravimetric analyses (TGA) measurement was performed under nitrogen with temperature increasing from 25 °C to 800 °C at a rate of 10 °C/min. Powder X-ray diffraction (PXRD) patterns were recorded on a Rigaku SmartLab X-ray diffractometer by setting powder on a glass substrate from $2\theta = 2.0°$ up to 30° with 0.01° increment. Nitrogen gas sorption curves were measured on a Micrometrics TriStar II Plus gas sorption instrument. Before measurement, powder samples were degassed in vacuum at 120 °C for 6 h. The Brunauer-Emmett-Teller (BET) approach was introduced to evaluate the surface areas.

### Time-resolved terahertz spectroscopy (TRTS)

The TRTS setup was powered by an amplified Ti: sapphire laser emitting ~ 50 fs laser pulses with a central wavelength of 800 nm and a repetition rate of 1 kHz. The fundamental 800 nm output is divided into three branches for optical excitation, THz generation, and THz detection[43]. The powder samples were sandwiched between two fused silica substrates and measured in a transmission geometry in a dry-nitrogen purged environment. For optical excitations, a 400 nm pump beam was obtained by frequency doubling of the 800 nm beam using a $\beta$-barium borate (BBO) crystal. A single-cycle THz probe pulse was generated via optical rectification by focusing 800 nm pulses onto a 1 mm thick ZnTe (110) crystal. The generated THz pulse was focused on the samples by a pair of off-axis parabolic mirrors, and the time-dependent THz electrical field was detected by the electro-optic sampling method.

### Computational details

Molecular modeling and Pawley refinement were carried out using Reflex, a software package for crystal determination from XRD pattern, implemented in MS modeling version 4.4 (Accelrys Inc.). Pawley refinement iteratively optimizes the lattice parameters until the $R_p$ and $R_{wp}$ values converge. Geometries of AA and AB COFs were calculated using Density Functional Tight Binding (DFTB) as implemented in DFTB+ version 20.1[52]. All atom pairs were described using standard parameters from the mio-0-1 parameter set[53]. Following geometry optimization, the electronic band structures and density of states (DOS) were calculated using the third-order DFTB and the 3ob-3-1 parameter set[54]. Using the electronic band structure calculated by the DFTB method, we derive the electron mass ($m_e$) and hole mass ($m_h$) by fitting the conduction band extrema and valence band extrema along a specific momentum path in the Brillouin zone with parabolic dispersion, respectively. The electron-hole reduced effective charge carrier mass ($m^*$) is then obtained using the Eq. (3):

$$\frac{1}{m^*} = \left[ \frac{1}{m_e} + \frac{1}{m_h} \right]^{-1} \tag{3}$$

All property calculations were undertaken in AMS-DFTB [AMS DFTB 2020, SCM, Theoretical Chemistry, Vrije Universiteit, Amsterdam, The Netherlands].

## Data availability

All data generated in this study are provided in the Supplementary Information/Source Data file. Source data are present. Source data are provided in this paper.

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

## Acknowledgements

This work was financially supported by the Max Planck Society, the National Natural Science Foundation of China (Grant 22371087 and 2237010919 to E.J.), the National Key Research and Development Program of China (2024YFB3815700 to E.J.), and the 111 centers (B17020 to E.J.). E.J. acknowledges support from the start-up grant and Open Project of the State Key Laboratory of Supramolecular Structure and Materials (sklssm2024012 to E.J.) of Jilin University. S.F. acknowledges the China Scholarship Council for financial support.

## Author contributions

E.J., K.M., and H.W. conceived and designed the project. S.F., X.L., G.W., and Y.G. contributed to the synthesis and characterizations of the synthesized COFs with the help of E.J., K.M., M.B., and H.W. M.A. contributed to the theoretical calculations. S.F., E.J., and H.W. co-wrote the manuscript, with contributions from all authors.

## Funding

## Competing interests

The authors declare no competing interests.
