## [Peer Review file · Nature Communications]

Dimensional Evolution of Charge Mobility and Porosity in Covalent Organic Frameworks

Corresponding Author: Professor Hai Wang

Version 0:

Reviewer comments:

Reviewer #1

(Remarks to the Author)

This manuscript by Wang and coworkers describes the transition from one-dimensional (1D) to two-dimensional (2D) covalent organic frameworks (COFs) and studies the effects of dimensional evolution on charge mobility and porosity. The authors show that the transition from 1D to 2D networks leads to a significant increase in surface area, along with a decrease in local charge mobility, due to substitution-induced electronic band flattening. The authors have provided solid data on a series of Pery-COFs. The conceptual novelty of the work is well articulated in the manuscript because developing COFs with both high porosity and excellent charge transport properties remains a significant challenge. However, we do have the following scientific concerns which should be addressed before publication in Nature Communications.

1. We are curious whether the design strategy of modulating dimensionality to balance porosity and charge mobility could be extended to other COFs, raising the possibility of the generality of the study. The authors have demonstrated how charge mobility can be enhanced by controlling substitution patterns. However, it remains unclear whether this approach (increasing dimensionality) can be applied to other COFs to achieve dual functionality of porosity and charge mobility. The authors should clarify this applicability or discuss the limitations of their design strategy.
2. The surface area of 947 m² g⁻¹ and charge mobility of 49 ± 10 cm² V⁻¹ s⁻¹ were obtained for 1.5D Pery-COF. We recommend comparing these values with those of state-of-the-art conductive COFs.
3. The Pawley refinement for the 1D Pery-COF suggested P1 space group, but the cell parameters (a = 39.3410 Å, b = 27.8420 Å, c = 3.5950 Å, α = β = γ = 90°) indicate an orthorhombic crystal system. The authors should clarify which classification is correct. In addition, the space group for the 1.5D Pery-COF should be confirmed, as the cell parameters (a = 21.7971 Å, b = 29.6484 Å, c = 3.8798 Å, α = β = γ = 90°) also suggest an orthorhombic crystal system, rather than the monoclinic P2/m space group.
4. A detailed procedure should be provided for the method used for electronic structure calculations with Materials Studio. Computational details should be provided for this study.
5. The full name of LUMO appears to be incorrect.
6. Although the authors did not explicitly indicate this, the 'D' in 1D and 2D Pery-COFs stands for dimensionality. In this regard, the term '1.5D Pery-COF' may be a bit confusing for the readers, as this COF also has a 2D network. It would be advisable to reconsider and revise the nomenclature for the series of COFs.

Reviewer #2

(Remarks to the Author)

Reviewer #3

(Remarks to the Author)

In the manuscript entitled "From One to Two Dimensions in Covalent Organic Frameworks – Evolution of Charge Mobility and Porosity" (NCOMMS-24-50919) by Fu et al., the authors present an innovative design of perylene-based COFs with different dimensionalities and substitution patterns. This structural control allows fine-tuning of the inherent porosity and electronic properties. By integrating surface area characterization, ultrafast terahertz spectroscopy, and theoretical calculations, the authors establish clear structure-property relationships and provide fundamental insights for designing COFs with both high porosity and charge mobility.

Overall, the manuscript is well written with a thorough discussion of the results. The COF design for the transport researches is very smart. The content is relevant to current research interests across multiple frontiers, including reticular chemistry, organic electronics, and catalysis, and it also aligns well the scope of Nature Communications. I recommend its publication in Nature Communications after addressing the following minor issues.

1. The achieved surface area and charge mobility should be thoroughly compared with state-of-the-art COFs to demonstrate their significance.
2. The authors should discuss the applicability of this synthetic strategy to other COF systems, especially high-porosity COFs and fully conjugated COFs.
3. What is the source of the error bars for the red histogram in Figure 3D? A description should be provided in the caption.

Reviewer #4

(Remarks to the Author)

Fu et al reported the development of 0-to-2 dimensional covalent organic frameworks (COFs) by varying the substituent of building unit. Especially, the authors performed the systematic time-resolved THz spectroscopy in order to estimate the charge carrier mobility related to the dimension of perylene-based COFs. The findings are novel and supported by the experimental results, and should be of interest of to the community working in COF and reticular chemistry. The manuscript merits its publication in Nature Communication, but the following comments should be addressed by the authors.

1) The authors categorized the dimension of the COFs in terms of the type of linker moieties between the perylene units and the degree of electronic coupling. Even though 1.5D Pery-COFs has a weak electronic coupling between 1D channels, the molecular structure of 1.5D Pery-COFs adopts the two-dimensional layered structure similar to that in 2D Pery-COFs. I think that the term of 1.5D COFs can highly confuse potential readers. Considering the previous studies for COFs, the dimension of COFs is defined in view of the molecular structure, not a degree of electronic coupling.

2) Based on the KM-reflectance absorption spectra for Pery-COFs, the simulated absorption spectra show the spectral red-shift of the absorption band compared to the experimental one. Especially, 1D Pery-COFs has a very low absorption feature around 700 nm (1.77 eV). I suspect that the DFTB method fails to describe the electronic transition in the Pery-COFs. Considering the HOMO-LUMO pictures, the 1.5D Pery-COFs shows the change of the electronic population from perylene core to linker moieties, implying the charge-transferred (CT) transition. In contrast, such a CT-like character is weak in the 1D Pery-COFs. I am wondering how the weak CT character in 1D Pery-COFs can cause the huge spectral red-shift compared to the other COFs.

3) To estimate the charge carrier mobility, the authors tried to fit the time-resolved THz spectra based on the DS model. For clarity, the authors should display all the fitting data including the real- and imaginary part.

4) The local mobility is highly related to the charge carrier time and effective charge mass. The charge carrier time originates from the kinetic component determined from the time-resolved THz spectroscopy. Then, I am wondering how the electron-hole effective mass (m_0) can be determined from the DFTB calculation. As I read the manuscript, I could not find the details for that. The author should provide all the computation details in the manuscript.

5) The direction of charge carrier flow is only discussed in the single sheet of COFs. In the XRD data, it is evident that all the Pery-COFs have the stacked AA structure, not the isolated single sheet. I suggest that the author should discuss the inter-layer charge flow and its effect in the observed charge mobility.

Version 1:

Reviewer comments:

Reviewer #1

(Remarks to the Author)

The authors have provided a thorough and comprehensive response to all of our queries. We are pleased to see that our comments have been adequately addressed, and the revisions have significantly improved the quality of the manuscript. We now have an additional comment. We suggest that the authors include charge mobility data for the newly introduced COF series. This data could further strengthen the paper by demonstrating how the dimensionality of these materials contributes to their dual functionality of porosity and charge mobility, thereby enhancing the overall impact and relevance of the study.

Reviewer #2

(Remarks to the Author)

Reviewer #3

(Remarks to the Author)

The response and the revision from the authors have fully addressed the concerns from the reviewer. It is recommended for publication in this current version.

Reviewer #4

(Remarks to the Author)

In this revised version, the authors have made strong arguments, providing precise explanation, and new data to support the discussion, in addition to adding better description of the data and figures. They well addressed the points raised in my previous comments on the initial manuscript. Especially, the added discussion about the charge mass would be helpful to understand the charge carrier mobility in COFs for the potential readers.

I recommend the publication of this manuscript as it is.

Reviewer 1:

This manuscript by Wang and coworkers describes the transition from one-dimensional (1D) to two-dimensional (2D) covalent organic frameworks (COFs) and studies the effects of dimensional evolution on charge mobility and porosity. The authors show that the transition from 1D to 2D networks leads to a significant increase in surface area, along with a decrease in local charge mobility, due to substitution-induced electronic band flattening. The authors have provided solid data on a series of Pery-COFs. The conceptual novelty of the work is well articulated in the manuscript because developing COFs with both high porosity and excellent charge transport properties remains a significant challenge. However, we do have the following scientific concerns which should be addressed before publication in Nature Communications.

Response: We are gratified that Reviewer 1 recognized the conceptual novelty of our study and appreciated our efforts in developing COFs with both high porosity and excellent charge transport properties. Below, we provide a point-by-point response to the reviewer's remaining minor comments.

Comment 1: We are curious whether the design strategy of modulating dimensionality to balance porosity and charge mobility could be extended to other COFs, raising the possibility of the generality of the study. The authors have demonstrated how charge mobility can be enhanced by controlling substitution patterns. However, it remains unclear whether this approach (increasing dimensionality) can be applied to other COFs to achieve dual functionality of porosity and charge mobility. The authors should clarify this applicability or discuss the limitations of their design strategy.

Response: This is an insightful comment. To evaluate the applicability of the design strategy, we made substantial efforts to synthesize three additional COFs with varying dimensionalities and substitution patterns based on pyrene units. Their structures were characterized using PXRD (**Fig. R1 a–c**), and their fundamental properties, including BET surface area (**Fig. R1 d–e**), FTIR (**Fig. R2**), TGA (**Fig. R3**) and SEM morphology (**Fig. R4**), were further comprehensively investigated. The three new pyrene-based COFs, namely 1D Py-COF, 2D PL-Py-COF, and 2D ML-Py-COF, were synthesized via Schiff-base condensation of 4,4',4'',4'''-(pyrene-1,3,6,8-tetrayl)tetraaniline (pyTTA) with TPDA, FTDA, FQDA, respectively, as shown in **Scheme R1**.

Scheme R1. Synthetic routes of pyrene-based COFs.

The synthesis details are as follows:

For 1D Py-COF: A pyrex tube (10 mL) was placed with 4,4',4'',4'''-(pyrene-1,3,6,8-tetrayl)tetraaniline (pyTTA) (10.0 mg, 0.018 mmol) and TPDA (10.1 mg, 0.035 mmol). A mixture of ODCB (0.20 mL)/*n*-BuOH (0.80 mL)/6 M HAc (0.10 mL) was added, followed by sonication for 5 min and degassing through three freeze-pump-thaw cycles. The tube was sealed under vacuum and heated at 120 °C for 72 h. After cooling to room temperature, the precipitate was centrifugated and washed three times with THF and acetone. Further purification was carried out by Soxhlet extraction in THF for 24 h and acetone for 24 h. The resulting powder was activated using supercritical carbon dioxide and dried under vacuum at 60 °C for 12 h, affording 1D Py-COF in a 53% isolated yield.

For 2D ML-Py-COF: A pyrex tube (10 mL) was placed with pyTTA (10.0 mg, 0.018 mmol) and FQDA (10.0 mg, 0.018 mmol). A mixture of ODCB (0.80 mL)/*n*-BuOH (0.20 mL)/6 M HAc (0.10 mL) was added, followed by sonication for 5 min and degassing through three freeze-pump-thaw cycles. The tube was sealed under vacuum and heated at 120 °C for 72 h. After cooling to room temperature, the precipitate was centrifuged and washed three times with THF and acetone. Further purification was carried out by Soxhlet extraction in THF for 24 h and acetone for 24 h. The resulting residue was activated using supercritical carbon dioxide and dried under vacuum at 60 °C for 12 h, affording 2D ML-Py-COF in a 69% isolated yield.

For 2D PL-Py-COF: A pyrex tube (10 mL) was placed with pyTTA (12.0 mg, 0.021 mmol) and FTDA (10.4 mg, 0.021 mmol). A mixture of ODCB (0.20 mL)/*n*-BuOH (0.80 mL)/6 M HAc (0.10 mL) was added, followed by sonication for 5 min and degassing through three freeze-pump-thaw cycles. The tube was sealed under vacuum and heated at 120 °C for 72 h. After cooling to room temperature, the precipitate was centrifuged and washed three times with THF and acetone. Further purification was carried out by Soxhlet extraction in THF for 24 h and acetone for 24 h. The resulting residue was activated using supercritical carbon dioxide and dried under vacuum at 60 °C for 12 h, affording 2D PL-Py-COF in a 66% isolated yield.

Fig. R1 a–c Powder X-ray diffraction (PXRD) of experimentally observed (purple curves), Pawley refined (green curves), their difference (blue curves), and simulated AA (black curves) patterns of (a) 1D Py-COF, (b) 2D PL-Py-COF, and (c) 2D ML-Py-COF; d–e Surface area and pore size distributions of (d) 1D Py-COF, (e) 2D PL-Py-COF, and (f) 2D ML-Py-COF.

Fig. R2 FTIR spectra of (a) 1D Py-COF, (b) 2D PL-Py-COF, and (c) 2D ML-Py-COF and the used monomers.

Fig. R3 TGA of 1D Py-COF (red curve), 2D ML-Py-COF (blue curve), and 2D PL-Py-COF (black curve).

Fig. R4 SEM images of (a) 1D Py-COF, (b) 2D PL-Py-COF, and (c) 2D ML-Py-COF.

Optical and THz photoconductivity characterizations

Fig. R5 (a) Solid-state UV/vis diffuse reflectance spectra. (b) Tauc plots showing the band gaps determined from the Kubelka-Munk-transformed (KM) reflectance spectra of 1D Py-COF (red curve), 2D ML-Py-COF (blue curve), and 2D PL-Py-COF (grey curve).

Fig. R6 (a) Time-resolved photoconductivity normalized by absorbed photon density ($\Delta\sigma/N_{abs}$) of Py-COFs. (b) Comparison of $\Delta\sigma/N_{abs}$ of Py-COFs. The error bars represent the standard deviation, estimated from point-to-point variation.

We have further investigated the optical and charge transport properties of pyrene-based COFs. In line with our results from Pery-COFs, we observe that increasing the dimensionality from 1D to 2D results in (i) an increase in bandgap (from 2.15 eV to 2.28 eV; see **Fig. R5**) and (ii) a

reduced photoconductivity amplitude per absorbed photon (see **Fig. R6**). Again, the new series of pyrene-based COFs demonstrated that the balance between surface area and charge mobility can be effectively optimized by judiciously tuning the dimensionality and substitution pattern, enhancing the applicability of the design strategy.

Action: For readability, we have discussed the applicability and limitations of the developed design strategy in the revised manuscript (below) and provided detailed characterization in the Supplementary Information (**Supplementary Figs. 11–16, Pages 17–22, and Supplementary Tables 7–9, Pages 37–42**).

*“...To evaluate the broader applicability of our synthetic strategy and observed trends, we extended our synthesis from the current Pery-based, to pyrene-based COFs with varying dimensionalities and substitution patterns, namely 1D Py-COF, 2D PL-Py-COF, and 2D ML-Py-COF, via Schiff-base condensation of 4,4',4'',4'''-(pyrene-1,3,6,8-tetrayl)tetraaniline (pyTTA) with TPDA, FTDA, FQDA, respectively (see details in **Supplementary Information**). Based on systematic structural, optical, and THz photoconductivity characterizations (**Supplementary Figs. 11–16 and Supplementary Tables 7–9**), we observe that increasing the dimension from 1D to 2D results in (i) an increase in bandgap (from 2.15 eV to 2.28 eV; **Supplementary Fig.14**) and (ii) a reduced photoconductivity amplitude per absorbed photon (**Supplementary Fig.15**). Being in line with our results from Pery-COFs, the new series of pyrene-based COFs demonstrated again that the balance between surface area and charge mobility can be effectively optimized by judiciously tuning the dimensionality and substitution pattern, enhancing the applicability of the design strategy....”*

The corresponding description has been added to the manuscript in **Lines 8–21, Page 13**.

Comment 2: The surface area of $947 \text{ m}^2 \text{ g}^{-1}$ and charge mobility of $49 \pm 10 \text{ cm}^2 \text{ V}^{-1} \text{ s}^{-1}$ were obtained for 1.5D Pery-COF. We recommend comparing these values with those of state-of-the-art conductive COFs.

Response and action: We fully agree with the reviewer that a comprehensive comparison of the surface areas and charge mobilities of state-of-the-art conductive COFs would underscore the importance of the COFs developed in this study and provide readers with a clearer understanding (**Table R1**). To this end, we have now summarized the surface areas and charge mobilities of state-of-the-art conductive COFs in **Supplementary Table 6, Pages 35–36**, and linked it in the revised manuscript in **Lines 28–29 and Lines 1–4, Pages 11–12** as follows: “...Knowing m^*

from DFT calculations (see computational details in **Methods** and calculated carrier masses in **Supplementary Table 5**), μ_{loc} of 1D Pery-COF, 2D ML-Pery-COF, and 2D PL-Pery-COF are estimated to be 66 ± 14 , 49 ± 10 , and $21 \pm 4 \text{ cm}^2 \cdot \text{V}^{-1} \cdot \text{s}^{-1}$, respectively. Utilizing the developed strategy, the achieved surface area and charge transport properties compare favorably with those of state-of-the-art conductive COFs (**Supplementary Table 6**)....”

Table R1. A comparison of surface areas and charge mobilities of conductive COFs.

Sample	τ (fs)	μ ($\text{cm}^2 \cdot \text{V}^{-1} \cdot \text{s}^{-1}$)	A ($\text{m}^2 \cdot \text{g}^{-1}$)	Ref
1D-Pery-COF	43 ± 4	66 ± 14^a	370	This work
2D ML- Pery-COF	43 ± 4	49 ± 10^a	947	This work
2D PL-Pery-COF	43 ± 4	21 ± 4^a	944	This work
1D Py-COF	N/A	N/A	321	This work
2D ML-Py-COF	N/A	N/A	947	This work
2D PL-Py-COF	N/A	N/A	666	This work
CuPc-MIDA-COF	23 ± 13	13.3 ± 7.5^a	442	7
CuPc-MIDA-COF	N/A	8.3^b	442	7
HHTP-MIDA-COF	16 ± 12	3.4 ± 2.5^a	1170	7
H ₂ P-COF	N/A	3.5^c	1894	8
CuP-COF	N/A	0.19^c	1713	8
ZnP-COF	N/A	0.048^c	1724	8
NiPc-COF	N/A	1.3^c	624	9
NiPc-BTDA COF	N/A	0.6^c	877	10
COF-366	N/A	8.1^c	735	11
COF-66	N/A	3^c	360	11
2D D-A COF	N/A	0.05^c	2021	12
CS-COF	N/A	4.2^c	776	13
TTF-Ph-COF	N/A	0.2^c	1014	14
TTF-Py-COF	N/A	0.08^c	817	14
HBC-COF (film)	N/A	0.7^c	965	15
CuPc-pz COF	N/A	0.9 ± 0.2^b	458.9	16

ZnPc-pz COF	N/A	4.8 ± 0.7^b	487.4	16
NiPc-CoTAA (film)	N/A	0.15^b	186	17
BUCT-COF-1	N/A	2.75 ± 0.22^b	976.6	18
DBOV-COF (film)	36 ± 6	0.6 ± 0.1^d	581	19
sp ² c-COF	41 ± 5	22.1 ± 2.7^d	N/A	20
sp ² c-COF-6	N/A	2.3 ± 0.5^d	667	21
sp ² c-COF-8	N/A	$< 0.1^d$	569	21
sp ² c-COF-9	N/A	5.8 ± 0.9^d	755	21
DHP-COF	28 ± 6	N/A	2031	22
c-HBC-COF	87 ± 5	44^d	1098	22
TPB-TFB COF (film)	72	165 ± 10^d	1232	23
V-2D-COF-W1	49 ± 8	1.4^d	245	24
V-2D-COF-W3	37 ± 6	10.3^d	118	24
V-2D-COF-W4	56 ± 8	0.6^d	333	24
2DPAV-BDT-BT	33 ± 4	65^d	N/A	25
2DPAV-BDT-BP	36 ± 5	17^d	N/A	25
2DCP-NiPc (film)	76 ± 3	971 ± 44^d	138	26
2DCP-CuPc (film)	45 ± 3	460 ± 31^d	191	26
AntTTH	N/A	~ 0.1	940	27

^aMobility obtained by time-resolved THz spectroscopy without taking into account the backscattering effect; ^bMobility obtained by Hall effect measurements; ^cMobility obtained by flash-photolysis time-resolved microwave conductivity; ^dMobility obtained by time-resolved THz spectroscopy taking into account the backscattering effect.

Comment 3: The Pawley refinement for the 1D Pery-COF suggested P1 space group, but the cell parameters ($a = 39.3410 \text{ \AA}$, $b = 27.8420 \text{ \AA}$, $c = 3.5950 \text{ \AA}$, $\alpha = \beta = \gamma = 90^\circ$) indicate an orthorhombic crystal system. The authors should clarify which classification is correct. In addition, the space group for the 1.5D Pery-COF should be confirmed, as the cell parameters ($a = 21.7971 \text{ \AA}$, $b = 29.6484 \text{ \AA}$, $c = 3.8798 \text{ \AA}$, $\alpha = \beta = \gamma = 90^\circ$) also suggest an orthorhombic crystal system, rather than the monoclinic P2/m space group.

Response: We thank the reviewer for this comment. After carefully re-evaluating our simulated crystalline structures, we confirm that the classification of the crystal cell of 1D Pery-COF is correct. The result is consistent with the output from the “find symmetry” function in Materials

Studio. The $P1$ space group can exist in any crystal system, including the orthorhombic system. But for the 2D ML-Pery-COF, the correct space group should be $P1$ instead of $P2/m$.

Action: We have corrected the corresponding description in the revised manuscript (**Line 19, Page 7**) and supporting information (**Page 31**).

Comment 4: A detailed procedure should be provided for the method used for electronic structure calculations with Materials Studio. Computational details should be provided for this study.

Response: We thank the reviewer for this suggestion. In fact, what we wanted to express here is “absorption features” rather than “electronic structures”. Below, we provide computational details of both absorption features and electronic structures.

Computational details of absorption: The simulation of absorption features was performed using DFT with the CASTEP package in Materials Studio. The Perdew-Burke-Ernzerhof (PBE) generalized gradient approximation (GGA) functional was employed to calculate the electronic exchange-correlation energies. The Grimme DFT-D correction was adopted for the van der Waals interactions due to the failure of the GGA/PBE functional to describe nonlocal dispersion forces. The cut-off energy of the plane wave basis set was set to 435 eV for geometry optimization and energy calculations.

Computational details of electronic structures: The electronic structure calculations were performed as follows: Geometries of monolayer, AA, AB, and slip-stacked 1D Pery-COF, 2D ML-Pery-COF, and 2D PL-Pery-COF were constructed using AuToGraFS (*J. Phys. Chem. A*, 2014, 118, 9607-9614) and 3rd order Density Functional Tight Binding (DFTB3). All atom pairs were described using standard parameters from the 3ob-3-1 parameter set, including D3-BJ treatment of dispersion. Cell parameters were optimized simultaneously with atom positions with no symmetry or constraints applied. After optimization, it was found that some unit cells had small deviations from 90° angles. These geometries were re-optimized with fixed cell angles. A reliable symmetric K-space was employed for all calculations. Band structure and effective masses were subsequently calculated using the same calculation parameters (*J. Chem. Theory Comput.*, 2013, 9, 338-354). All calculations were undertaken using AMS2021 (AMS 2022.1, SCM, Theoretical Chemistry, Vrije Universiteit, Amsterdam, The Netherlands, <http://www.scm.com>). HOMO and LUMO orbitals were visualized from the optimized structures using waveplot 0.2.

Action: We have corrected the corresponding description in the revised manuscript (**Lines 4–6, Page 9**) as follows: “...To understand the increase in bandgap with dimensionality, *absorption features* of 1D Pery-COF, 2D ML-Pery-COF, and 2D PL-Pery-COF were *simulated* using Materials Studio (version 4.4, *see details in Supplementary Information*)⁴⁸....”. Furthermore, we have provided the computational details of both “absorption features” and “electronic structures” in the **Page 6, Supplementary Information**.

Comment 5: The full name of LUMO appears to be incorrect.

Response and action: We thank the reviewer for pointing this out. The typo has been corrected in the revised manuscript: “The spatial extent of the highest occupied molecular orbital (HOMO) and the lowest *unoccupied* molecular orbital (LUMO) of 1D Pery-COF, 2D ML-Pery-COF, and 2D PL-Pery-COF were calculated using the periodic DFTB structures (**Supplementary Table 5**).” in **Lines 14–17, Page 9**.

Comment 6: Although the authors did not explicitly indicate this, the ‘D’ in 1D and 2D Pery-COFs stands for dimensionality. In this regard, the term ‘1.5D Pery-COF’ may be a bit confusing for the readers, as this COF also has a 2D network. It would be advisable to reconsider and revise the nomenclature for the series of COFs.

Response and action: We fully understand the reviewer’s concern. Following this comment, we have revised the nomenclature of this series of COFs from 1D, 1.5D, and 2D perylene-based COF to 1D Pery-COF, 2D ML-Pery-COF, and 2D PL-Pery-COF.

Reviewer 2:

Response: We appreciate the reviewer for his/her time and effort in co-reviewing this manuscript and providing valuable comments.

Reviewer 3:

In the manuscript entitled “From One to Two Dimensions in Covalent Organic Frameworks – Evolution of Charge Mobility and Porosity” (NCOMMS-24-50919) by Fu et al., the authors present an innovative design of perylene-based COFs with different dimensionalities and substitution patterns. This structural control allows fine-tuning of the inherent porosity and electronic properties. By integrating surface area characterization, ultrafast terahertz spectroscopy, and theoretical calculations, the authors establish clear structure-property relationships and provide fundamental insights for designing COFs with both high porosity and charge mobility. Overall, the manuscript is well written with a thorough discussion of the results. The COF design for the transport researches is very smart. The content is relevant to current research interests across multiple frontiers, including reticular chemistry, organic electronics, and catalysis, and it also aligns well the scope of *Nature Communications*. I recommend its publication in Nature Communications after addressing the following minor issues.

Response: We greatly appreciate the positive feedback from reviewer 3 and are thrilled by her/his recognition of the structural design and multidisciplinary application potential of our work. In the following, we provide a point-by-point response to the minor revisions suggested by the reviewer.

Comment 1: The achieved surface area and charge mobility should be thoroughly compared with state-of-the-art COFs to demonstrate their significance.

Response and action: We fully agree with the reviewer that a comprehensive comparison of the surface areas and charge mobilities of state-of-the-art conductive COFs would underscore the importance of the COFs developed in this study and provide readers with a clearer understanding (**Table R1**). To this end, we have now summarized the surface areas and charge mobilities of state-of-the-art conductive COFs in **Supplementary Table 6, Pages 35–36**, and linked it in the revised manuscript in **Lines 28–29** and **Lines 1–4, Pages 11–12** as follows: “...Knowing m^* from DFT calculations (see computational details in *Methods* and calculated carrier masses in **Supplementary Table 5**), μ_{loc} of 1D Pery-COF, 2D ML-Pery-COF, and 2D PL-Pery-COF are estimated to be 66 ± 14 , 49 ± 10 , and 21 ± 4 $\text{cm}^2 \cdot \text{V}^{-1} \cdot \text{s}^{-1}$, respectively. Utilizing the developed strategy, the achieved surface area and charge transport properties compare favorably with those of state-of-the-art conductive COFs (**Supplementary Table 6**)....”

Table R1. A comparison of surface areas and charge mobilities of conductive COFs.

Sample	τ (fs)	μ ($\text{cm}^2 \cdot \text{V}^{-1} \cdot \text{s}^{-1}$)	A ($\text{m}^2 \cdot \text{g}^{-1}$)	Ref
1D-Pery-COF	43 ± 4	$66 \pm 14^{\text{a}}$	370	This work
2D ML- Pery-COF	43 ± 4	$49 \pm 10^{\text{a}}$	947	This work
2D PL-Pery-COF	43 ± 4	$21 \pm 4^{\text{a}}$	944	This work
1D Py-COF	N/A	N/A	321	This work
2D ML-Py-COF	N/A	N/A	947	This work
2D PL-Py-COF	N/A	N/A	666	This work
CuPc-MIDA-COF	23 ± 13	$13.3 \pm 7.5^{\text{a}}$	442	7
CuPc-MIDA-COF	N/A	8.3^{b}	442	7
HHTP-MIDA-COF	16 ± 12	$3.4 \pm 2.5^{\text{a}}$	1170	7
H ₂ P-COF	N/A	3.5^{c}	1894	8
CuP-COF	N/A	0.19^{c}	1713	8
ZnP-COF	N/A	0.048^{c}	1724	8
NiPc-COF	N/A	1.3^{c}	624	9
NiPc-BTDA COF	N/A	0.6^{c}	877	10
COF-366	N/A	8.1^{c}	735	11
COF-66	N/A	3^{c}	360	11
2D D-A COF	N/A	0.05^{c}	2021	12
CS-COF	N/A	4.2^{c}	776	13
TTF-Ph-COF	N/A	0.2^{c}	1014	14
TTF-Py-COF	N/A	0.08^{c}	817	14
HBC-COF (film)	N/A	0.7^{c}	965	15
CuPc-pz COF	N/A	$0.9 \pm 0.2^{\text{b}}$	458.9	16
ZnPc-pz COF	N/A	$4.8 \pm 0.7^{\text{b}}$	487.4	16
NiPc-CoTAA (film)	N/A	0.15^{b}	186	17
BUCT-COF-1	N/A	$2.75 \pm 0.22^{\text{b}}$	976.6	18
DBOV-COF (film)	36 ± 6	$0.6 \pm 0.1^{\text{d}}$	581	19
sp ² c-COF	41 ± 5	$22.1 \pm 2.7^{\text{d}}$	N/A	20
sp ² c-COF-6	N/A	$2.3 \pm 0.5^{\text{d}}$	667	21

sp ² c-COF-8	N/A	< 0.1 ^d	569	21
sp ² c-COF-9	N/A	5.8 ± 0.9 ^d	755	21
DHP-COF	28 ± 6	N/A	2031	22
c-HBC-COF	87 ± 5	44 ^d	1098	22
TPB-TFB COF (film)	72	165 ± 10 ^d	1232	23
V-2D-COF-W1	49 ± 8	1.4 ^d	245	24
V-2D-COF-W3	37 ± 6	10.3 ^d	118	24
V-2D-COF-W4	56 ± 8	0.6 ^d	333	24
2DPAV-BDT-BT	33 ± 4	65 ^d	N/A	25
2DPAV-BDT-BP	36 ± 5	17 ^d	N/A	25
2DCP-NiPc (film)	76 ± 3	971 ± 44 ^d	138	26
2DCP-CuPc (film)	45 ± 3	460 ± 31 ^d	191	26
AntTTH	N/A	~0.1	940	27

^aMobility obtained by time-resolved THz spectroscopy without taking into account the backscattering effect; ^bMobility obtained by Hall effect measurements; ^cMobility obtained by flash-photolysis time-resolved microwave conductivity; ^dMobility obtained by time-resolved THz spectroscopy taking into account the backscattering effect.

Comment 2: The authors should discuss the applicability of this synthetic strategy to other COF systems, especially high-porosity COFs and fully conjugated COFs.

Response: To evaluate the applicability of the design strategy, we made substantial efforts to synthesize three additional COFs with varying dimensionalities and substitution patterns based on pyrene units. Their structures were characterized using PXRD (**Fig. R1 a–c**), and their fundamental properties, including BET surface area (**Fig. R1 d–e**), FTIR (**Fig. R2**), TGA (**Fig. R3**) and SEM morphology (**Fig. R4**), were further comprehensively investigated. The three new pyrene-based COFs, namely 1D Py-COF, 2D PL-Py-COF, and 2D ML-Py-COF, were synthesized via Schiff-base condensation of 4,4',4'',4'''-(pyrene-1,3,6,8-tetrayl)tetraaniline (pyTTA) with TPDA, FTDA, FQDA, respectively, as shown in **Scheme R1**.

Scheme R1. Synthetic routes of pyrene-based COFs.

The synthesis details are as follows:

For 1D Py-COF: A pyrex tube (10 mL) was placed with 4,4',4'',4'''-(pyrene-1,3,6,8-tetrayl)tetraaniline (pyTTA) (10.0 mg, 0.018 mmol) and TPDA (10.1 mg, 0.035 mmol). A mixture of ODCB (0.20 mL)/*n*-BuOH (0.80 mL)/6 M HAc (0.10 mL) was added, followed by sonication for 5 min and degassing through three freeze-pump-thaw cycles. The tube was sealed under vacuum and heated at 120 °C for 72 h. After cooling to room temperature, the precipitate was centrifugated and washed three times with THF and acetone. Further purification was carried out by Soxhlet extraction in THF for 24 h and acetone for 24 h. The resulting powder was

activated using supercritical carbon dioxide and dried under vacuum at 60 °C for 12 h, affording 1D Py-COF in a 53% isolated yield.

For 2D ML-Py-COF: A pyrex tube (10 mL) was placed with pyTTA (10.0 mg, 0.018 mmol) and FQDA (10.0 mg, 0.018 mmol). A mixture of ODCB (0.80 mL)/*n*-BuOH (0.20 mL)/6 M HAc (0.10 mL) was added, followed by sonication for 5 min and degassing through three freeze-pump-thaw cycles. The tube was sealed under vacuum and heated at 120 °C for 72 h. After cooling to room temperature, the precipitate was centrifugated and washed three times with THF and acetone. Further purification was carried out by Soxhlet extraction in THF for 24 h and acetone for 24 h. The resulting residue was activated using supercritical carbon dioxide and dried under vacuum at 60 °C for 12 h, affording 2D ML-Py-COF in a 69% isolated yield.

For 2D PL-Py-COF: A pyrex tube (10 mL) was placed with pyTTA (12.0 mg, 0.021 mmol) and FTDA (10.4 mg, 0.021 mmol). A mixture of ODCB (0.20 mL)/*n*-BuOH (0.80 mL)/6 M HAc (0.10 mL) was added, followed by sonication for 5 min and degassing through three freeze-pump-thaw cycles. The tube was sealed under vacuum and heated at 120 °C for 72 h. After cooling to room temperature, the precipitate was centrifugated and washed three times with THF and acetone. Further purification was carried out by Soxhlet extraction in THF for 24 h and acetone for 24 h. The resulting residue was activated using supercritical carbon dioxide and dried under vacuum at 60 °C for 12 h, affording 2D PL-Py-COF in a 66% isolated yield.

Fig. R1 a–c Powder X-ray diffraction (PXRD) of experimentally observed (purple curves), Pawley refined (green curves), their difference (blue curves), and simulated AA (black curves) patterns of (a) 1D Py-COF, (b) 2D PL-Py-COF, and (c) 2D ML-Py-COF; d–e Surface area and pore size distributions of (d) 1D Py-COF, (e) 2D PL-Py-COF, and (f) 2D ML-Py-COF.

Fig. R2 FTIR spectra of (a) 1D Py-COF, (b) 2D PL-Py-COF, and (c) 2D ML-Py-COF and the used monomers.

Fig. R3 TGA of 1D Py-COF (red curve), 2D ML-Py-COF (blue curve), and 2D PL-Py-COF (black curve).

Fig. R4 SEM images of (a) 1D Py-COF, (b) 2D PL-Py-COF, and (c) 2D ML-Py-COF.

Optical and conductivity characterizations

Fig. R5 (a) Solid-state UV/vis diffuse reflectance spectra. (b) Tauc plots showing the band gaps determined from the Kubelka-Munk-transformed (KM) reflectance spectra of 1D Py-COF (red curve), 2D ML-Py-COF (blue curve), and 2D PL-Py-COF (grey curve).

Fig. R6 (a) Time-resolved photoconductivity normalized by absorbed photon density ($\Delta\sigma_s/N_{abs}$) of Py-COFs. (b) Comparison of $\Delta\sigma_s/N_{abs}$ of Py-COFs. The error bars represent the standard deviation, estimated from point-to-point variation.

We have further investigated the optical and charge transport properties of pyrene-based COFs. In line with our results from Pery-COFs, we observe that increasing the dimensionality from 1D to 2D results in (i) an increase in bandgap (from 2.15 eV to 2.28 eV; see Fig. R5) and (ii) a reduced photoconductivity amplitude per absorbed photon (see Fig. R6). Again, the new series of pyrene-based COFs demonstrated that the balance between surface area and charge mobility can be effectively optimized by judiciously tuning the dimensionality and substitution pattern, enhancing the applicability of the design strategy.

Action: For readability, we have discussed the applicability and limitations of the developed design strategy in the revised manuscript (below) and provided detailed characterization in the

Supplementary Information (**Supplementary Figs.11–16, Pages 17–22, and Supplementary Tables 7–9, Pages 37–42**).

*“...To evaluate the broader applicability of our synthetic strategy and observed trends, we synthesized a series of pyrene-based COFs with varying dimensionalities and substitution patterns, namely 1D Py-COF, 2D PL-Py-COF, and 2D ML-Py-COF, via Schiff-base condensation of 4,4',4'',4'''-(pyrene-1,3,6,8-tetra-yl)tetraaniline (pyTTA) with TPDA, FTDA, FQDA, respectively (see details in **Supplementary Information**). Based on systematic structural, optical, and THz photoconductivity characterizations (**Supplementary Figs. 11–16**), we observe that increasing the dimension from 1D to 2D results in (i) an increase in bandgap (from 2.15 eV to 2.28 eV; **Supplementary Fig.14**) and (ii) a reduced photoconductivity amplitude per absorbed photon (**Supplementary Fig.15**). Being in line with our results from Pery-COFs, the new series of pyrene-based COFs demonstrated again that the balance between surface area and charge mobility can be effectively optimized by judiciously tuning the dimensionality and substitution pattern, enhancing the applicability of the design strategy....”*

The corresponding description has been added to the manuscript in **Lines 8–20, Page 13**.

Comment 3: What is the source of the error bars for the red histogram in Figure 3D? A description should be provided in the caption.

Response and action: The error bars of the red histogram in **Fig. 3d** represent the standard deviation, estimated from point-to-point variation. Specifically, to estimate the point-to-point variation, we measured up to five points on a single sample and derived a standard deviation of ~15% (which we applied across all groups). Following this comment, we also examined other potential error sources and confirmed that point-to-point variation is the primary error source in our experiments. To clarify this, we have added the following description to the caption (**Lines 13–14, Page 10**):

*“**d** Correlation between the relative mobility ($\Delta\sigma/N_{abs}$) of perylene-based COFs and the inverse of their electron-hole reduced effective masses ($1/m^*$). The error bars represent the standard deviation, estimated from point-to-point variation.”*

Reviewer: 4

Fu et al reported the development of 0-to-2 dimensional covalent organic frameworks (COFs) by varying the substituent of building unit. Especially, the authors performed the systematic time-resolved THz spectroscopy in order to estimate the charge carrier mobility related to the dimension of perylene-based COFs. The findings are novel and supported by the experimental results, and should be of interest of to the community working in COF and reticular chemistry. The manuscript merits its publication in Nature Communication, but the following comments should be addressed by the authors.

Response: We are pleased that Reviewer 4 recognized the novelty of our study and its potential significance for the COF and reticular chemistry communities, recommending it for publication in *Nature Communications* after revision. Below, we address the reviewer's remaining minor comments point by point.

Comment 1: The authors categorized the dimension of the COFs in terms of the type of linker moieties between the perylene units and the degree of electronic coupling. Even though 1.5D Pery-COFs has a weak electronic coupling between 1D channels, the molecular structure of 1.5D Pery-COFs adopts the two-dimensional layered structure similar to that in 2D Pery-COFs. I think that the term of 1.5D COFs can highly confuse potential readers. Considering the previous studies for COFs, the dimension of COFs is defined in view of the molecular structure, not a degree of electronic coupling.

Response and action: We fully understand the reviewer's concern. Following this comment, we have revised the nomenclature of this series of COFs from 1D, 1.5D, and 2D perylene-based COF to 1D Pery-COF, 2D ML-Pery-COF, and 2D PL-Pery-COF.

Comment 2: Based on the KM-reflectance absorption spectra for Pery-COFs, the simulated absorption spectra show the spectral red-shift of the absorption band compared to the experimental one. Especially, 1D Pery-COFs has a very low absorption feature around 700 nm (1.77 eV). I suspect that the DFTB method fails to describe the electronic transition in the Pery-COFs. Considering the HOMO-LUMO pictures, the 1.5D Pery-COFs shows the change of the electronic population from perylene core to linker moieties, implying the charge-transferred (CT) transition. In contrast, such a CT-like character is weak in the 1D Pery-COFs. I am wondering

how the weak CT character in 1D Pery-COFs can cause the huge spectral red-shift compared to the other COFs.

Response and action: We thank the reviewer for the insightful comments regarding (i) the discrepancy between calculated and experimental bandgaps and (ii) the relationship between CT-like character and spectral red-shift. For (i), it is well established that band gaps calculated via density functional theory using local or gradient-corrected exchange-correlation potentials are typically underestimated compared to experimental values (*J. Phys. Chem. C*, 2017, 121, 18862–18866). Despite this discrepancy in absolute values, the observed red-shift trend with decreasing dimensionality in our calculations is in good agreement with the experimental results. For (ii), we agree with the reviewer that the CT-like character should be weaker in the 1D Pery-COF compared to the other two 2D COFs. Nevertheless, the more pronounced red-shift observed in the 1D Pery-COF can be rationalized by its extended π -conjugation along the polymeric skeleton. As shown in **Fig. R7**, the twist angle between two adjacent benzene rings in the TPDA unit of 1D COF is 24.12° , significantly smaller than the 40.66° observed in the FTDA unit of the 2D COF. This larger twist angle in the 2D COF, resulting from stronger steric repulsion between the hydrogens of the para-substituted benzene rings, likely induces a more non-planar configuration, which hinders π -conjugation, increases the band gap, and results in a more pronounced blue-shift in its absorption feature. Notably, a similar trend is also observed in the newly synthesized series of pyrene-based COFs (**Fig. R2**), further corroborating our explanation. The corresponding description, “*The bandgaps of COFs are primarily determined by the extent of π -conjugation within their polymeric skeletons. In the 2D PL-Pery-COF, the twist angle between two adjacent benzene rings in the FTDA unit is 40.66° , larger than the 24.12° twist angle between the meta-substituted benzene and the central benzene ring in the TPDA unit of 1D Pery-COF (Supplementary Fig. 9). Consequently, the increased non-planar arrangement of the FTDA unit within the 2D PL-Pery-COF restricts the extension of π -conjugation, which likely accounts for the larger bandgap and the blue-shifted absorption spectra observed when compared to 1D COFs.*” has been added to the revised manuscript in the **Lines 7–14, Page 9**.

Fig. R7. Molecular configurations of (a) 2D PL-Pery-COF and (b) 1D Pery-COF.

Fig. R2 (a) Solid-state UV/vis diffuse reflectance spectra. (b) Tauc plots showing the band gaps determined from the Kubelka-Munk-transformed (KM) reflectance spectra of 1D Py-COF (red curve), 2D ML-Py-COF (blue curve), and 2D PL-Py-COF (black curve).

Comment 3: To estimate the charge carrier mobility, the authors tried to fit the time-resolved THz spectra based on the DS model. For clarity, the authors should display all the fitting data including the real- and imaginary part.

Response and action: Indeed, the fitting data in Fig. 3c includes both the real and imaginary parts, represented by the solid and dashed orange lines, respectively. We speculate that the misunderstanding may stem from the color or line style not being sufficiently clear. To improve visualization, we have modified the color and the line style as follows:

Fig. 3c Frequency-resolved photoconductivity ($\Delta\sigma(\omega)$) of perylene-based COFs measured at 0.5 ps after the maximum photoconductivity. The filled and empty circles represent the real and imaginary components of the complex THz photoconductivity of **1D Pery-COF** (red), **2D ML-Pery-COF** (blue, rescaled by $1.24\times$), and **2D PL-Pery-COF** (black, rescaled by $2.09\times$). The **red and purple** solid lines correspond to the global fitting of the real and imaginary components, respectively, of $\Delta\sigma(\omega)$ following the Drude-Smith (DS) model.

We have updated the caption in the manuscript in **Lines 5–11, Page 10**.

Comment 4: The local mobility is highly related to the charge carrier time and effective charge mass. The charge carrier time originates from the kinetic component determined from the time-resolved THz spectroscopy. Then, I am wondering how the electron-hole effective mass (m_0) can be determined from the DFTB calculation. As I read the manuscript, I could not find the details for that. The author should provide all the computation details in the manuscript.

Response and action: We thank the reviewer for this suggestion. To enhance clarity, we have included all relevant computational details in the **Methods** section and linked the corresponding text in the **Results** section in **Lines 23–25, Page 9, and Lines 12–18, Page 15**.

Results

“...These results indicate that (i) 1D Pery-COF forms directional charge-conducting channels with a relatively small electron-hole reduced effective mass (m^* , see computational details in **Methods**);...”

Computational details

“...Following geometry optimization, the electronic band structures and density of states (DOS) were calculated using the third-order DFTB and the 3ob-3-1 parameter set⁵⁴. Using the electronic band structure calculated by the DFTB method, we derive the electron mass (m_e) and

hole mass (m_h) by fitting the conduction band extrema and valence band extrema along a specific momentum path in the Brillouin zone with parabolic dispersion, respectively. The electron-hole reduced effective charge carrier mass (m^*) is then obtained using the relationship

$$\frac{1}{m^*} = \left[\frac{1}{m_e} + \frac{1}{m_h} \right]^{-1} \dots$$

Comment 5: The direction of charge carrier flow is only discussed in the single sheet of COFs. In the XRD data, it is evident that all the Pery-COFs have the stacked AA structure, not the isolated single sheet. I suggest that the author should discuss the inter-layer charge flow and its effect in the observed charge mobility.

Response: We appreciate the insightful comment regarding the direction of charge carrier flow. The THz pulse characterizes charge transport perpendicular to its propagation direction. Given the random orientation of the COF samples, the derived charge transport properties represent an effective value that, in principle, includes contributions from both in-plane (intra-layer) and out-of-plane (inter-layer) directions. While the current experimental paradigm and analytical protocol limit the ability to isolate directional charge transport properties, we agree with the reviewer that a discussion on directional charge transport is important. To this end, we have calculated the in-plane and out-of-plane electron and hole masses. A summary of the in-plane electron mass, out-of-plane electron mass, and hole mass is now provided in **Table R2**. Our findings reveal that: (i) 1D Pery-COF exhibits low in-plane and out-of-plane electron masses as well as a low hole mass, consistent with its highest charge mobility among the Pery-COF series; (ii) the most pronounced band-flattening effect observed in 2D PL-Pery-COF results from a combined effect of the substantial increase in hole mass and out-of-plane electron mass; and (iii) the high charge mobility maintained in 2D-ML Pery-COF relative to 1D Pery-COF is driven by a subtle balance between the increase in hole and in-plane electron mass and the decrease in out-of-plane electron mass. We have updated the **Supplementary Table 5** in **Page 34** in **supporting information**.

Table R2. DFTB calculated LUMO and HOMO of AA stacked 1D Pery-COF, 2D ML-Pery-COF, and 2D PL-Pery-COF.

	LUMO (eV)	HOMO (eV)	Electron mass (m_0)	Hole mass (m_0)

1D Pery-COF	-3.01	-4.42	In-plane 2.867 Out-of-plane 1.929	In-plane 2.211
2D ML-Pery-COF	-3.23	-4.95	In-plane 4.066 Out-of-plane 1.212	In-plane 5.182
2D PL-Pery-COF	-3.37	-5.04	In-plane 2.752 Out-of-plane 7.427	In-plane 16.237

Note that we present only in-plane hole masses. Out-of-plane hole masses are not considered due to their significantly larger values, which indicate unfavorable out-of-plane hole transport.

We have added these findings to the revised manuscript to provide more insights in **Lines 28–29, Page 11, and Lines 1–13, Page 12**, as follows:

“...Knowing m^* from DFT calculations (see computational details in **Methods** and calculated carrier masses in **Supplementary Table 5**), μ_{loc} of 1D Pery-COF, 2D ML-Pery-COF, and 2D PL-Pery-COF are estimated to be 66 ± 14 , 49 ± 10 , and 21 ± 4 $\text{cm}^2 \cdot \text{V}^{-1} \cdot \text{s}^{-1}$, respectively. Utilizing the developed strategy, the achieved surface area and charge transport properties compare favorably with those of state-of-the-art conductive COFs (**Supplementary Table 6**). While the current experimental paradigm and analytical protocol limit the ability to isolate directional charge transport properties, theoretical calculations of directional carrier masses suggest that: (i) 1D Pery-COF exhibits low in-plane and out-of-plane electron masses as well as a low hole mass, consistent with its highest charge mobility among the Pery-COF series; (ii) the most pronounced band-flattening effect observed in 2D PL-Pery-COF results from a combined effect of the substantial increase in hole mass and out-of-plane electron mass; and (iii) the high charge mobility maintained in 2D ML-Pery-COF relative to 1D Pery-COF is driven by a subtle balance between the increase in hole and in-plane electron mass and the decrease in out-of-plane electron mass. ...”

Reviewer 1: The authors have provided a thorough and comprehensive response to all of our queries. We are pleased to see that our comments have been adequately addressed, and the revisions have significantly improved the quality of the manuscript. We now have an additional comment. We suggest that the authors include charge mobility data for the newly introduced COF series. This data could further strengthen the paper by demonstrating how the dimensionality of these materials contributes to their dual functionality of porosity and charge mobility, thereby enhancing the overall impact and relevance of the study.

Response: We are gratified that Reviewer 1 is satisfied with our revisions and recognized the significant improvement of the manuscript. Following this comment, we have included the charge transport properties of the newly introduced COF series in Fig. 5 as follows:

Fig. 5 | Synthesis and charge transport properties of pyrene-based COF. a Synthetic routes of 1D Py-COF, 2D-ML-Py-COF, and 2D-PL-Py-COF. b Time-resolved photoconductivity normalized by absorbed photon density ($\Delta\sigma N_{abs}$) of pyrene-based COFs. c Comparison of the peak photoconductivity normalized by absorbed photon density ($\Delta\sigma N_{abs}$) of pyrene-based COFs. The error bars represent the standard deviation, estimated from point-to-point variation.

a**b****c**